# Earthquake Monitoring using Deep Learning: a case study on the Kahramanmaras Turkey Earthquake Aftershock Sequence

Wei Li[1], Megha Chakraborty[1,2], Jonas Köhler[1,2], Claudia Quinteros-Cartaya[1], Georg Rümpker[1,2], and Nishtha Srivastava[1,2]

[1]Frankfurt Institute for Advanced Studies, 60438 Frankfurt am Main, Germany
[2]Institute of Geosciences, Goethe-University, 60438 Frankfurt am Main, Germany

**Correspondence:** Nishtha Srivastava (srivastava@fias.uni-frankfurt.de)

**Abstract.** Seismic phase picking and magnitude estimation are fundamental aspects of earthquake monitoring and seismic event analysis. Accurate phase picking allows for precise characterization of seismic wave arrivals, contributing to a better understanding of earthquake events. Likewise, accurate magnitude estimation provides crucial information about an earthquake's size and potential impact. Together, these components enhance our ability to monitor seismic activity effectively. In this study, we explore the application of deep learning techniques for earthquake detection and magnitude estimation using continuous seismic recordings. Our approach introduces DynaPicker, which leverages dynamic convolutional neural networks to detect seismic body wave phases in continuous seismic data. We demonstrate DynaPicker's effectiveness using various open-source seismic datasets, including both window-format and continuous recordings. We evaluate its performance in seismic phase identification and arrival-time picking, as well as its robustness in classifying seismic phases using low-magnitude seismic data in the presence of noise. Furthermore, we integrate the phase arrival time information into a previously published deep-learning model for magnitude estimation. We apply this workflow to continuous recordings of aftershock sequences following the Turkey earthquake. The results of this case study showcase the reliability of our approach in earthquake detection, phase picking, and magnitude estimation, contributing valuable insights to seismic event analysis.

## 1 Introduction

Seismic phase picking, which plays an essential role in earthquake location identification and body-wave travel time tomography, is often performed manually. In order to achieve adequately automated seismic phase picking, many conventional approaches have been studied over the past few decades. Common algorithms developed for seismic phase picking include short-time average/long-time average (STA/LTA) (Allen, 1978) and Akaike information criterion (AIC) (Leonard and Kennett, 1999). The STA/LTA is mathematically formulated as the ratio of the average amplitude over a short time window to the average amplitude over a long time window. In STA/LTA, an event is detected when the ratio is greater than the defined threshold. The AIC solution is subject to the assumption that the seismogram can be split into auto-regressive (AR) segments, where the minimum AIC value is usually defined as the arrival time. However, neither STA/LTA nor AIC can achieve satisfactory performance for low signal-to-ratio (SNR) signals.

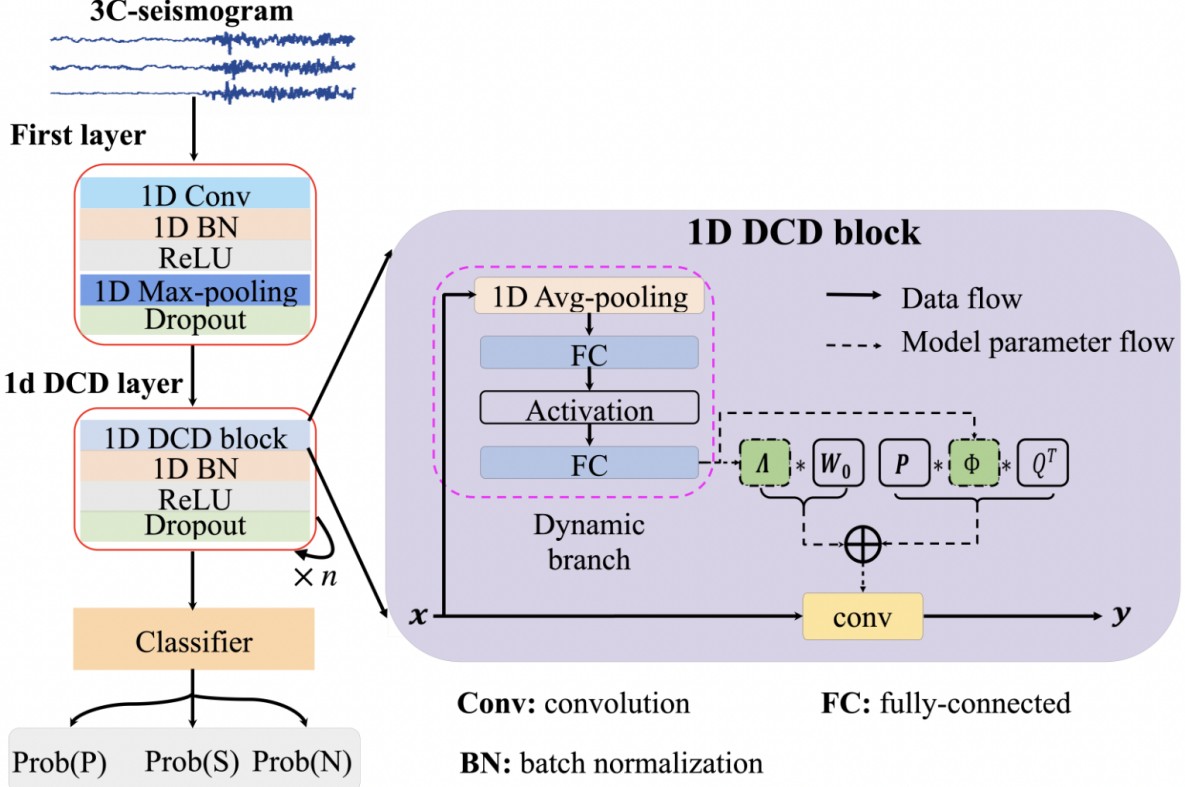

**Figure 1.** Schematic diagram for the proposed Dynamic Convolution Decomposition (DCD) based model. The model architecture represented on the left side includes a convolutional layer, 1D-DCD layers, and the classifier. The 1D-DCD block displayed on the right side is the backbone of the 1D-DCD layer, which is adapted from the work of Li et al. (2021b) and converted into the 1D case in this study. In a 1D-DCD block, the input $x$ first goes through a dynamic branch to generate $\mathbf{\Lambda}(x)$ and $\mathbf{\Phi}(x)$, and then to produce the convolution matrix $\boldsymbol{W}(x)$ using Eq. (3).

The past decades have witnessed a sharp increase in the amount of available seismic data owing to the advancement of seismic equipment and the expansion of seismic monitoring networks. This has increased the demand for a robust seismic phase picking method to deal with large volumes of seismic data. Deep learning has the merit of facilitating the processing of large amounts of data and extracting its features which makes it successful in diverse areas, especially in image processing (LeCun et al., 2015). The implementation of seismic phase picking can be considered similar to object identification in computer vision. Thus, the use of deep learning has been widely embraced in first-motion polarities identification of earthquake waveforms (Chakraborty et al., 2022a), seismic event detection (Perol et al., 2018; Mousavi et al., 2019b; Fenner et al., 2022; Li et al., 2022b), earthquake magnitude classification and estimation (Chakraborty et al., 2021, 2022b, c), and seismic phase picking (Ross et al., 2018; Zhu and Beroza, 2019; Mousavi et al., 2020; Li et al., 2021a, 2022a). Stepnov et al. (2021) stated that seismic phase picking approaches can be roughly divided into two main streams, continuous seismic waveform-based and

small window-format-based methods. The former is to process continuous seismic waveforms like earthquake-length windows of fixed duration with more complex triggers. The output of this type of model is the probability distribution over the fixed window length. The latter is to split the seismic waveform into small windows (e.g., 4 – 6 s (Ross et al., 2018)), where only one centered pick or noise is included. Then, each window is identified as one of three classes: P-wave, S-wave, and noise. Stepnov et al. (2021) concluded that for the former scenarios, those models can work well when scanning archives, whereas it is only suitable for pre-recorded data processing because of the restriction imposed by the required input window length. On the contrary, considering that the ground motion data are constantly received in small chunks, small windows allow the processing can be subsequently adapted to real-time monitoring as well. (Stepnov et al., 2021). As a result, the length of the long waveform can be formed by sequentially adding the successive chunk to the previous continuous data, and each chunk could be directly fed into the pre-trained model for class identification.

Most deep learning-based seismic phase classification model architectures largely rely on convolutional neural networks (CNN). CNN is capable of extracting meaningful features from the input data, which enables the neural network to achieve a good performance. However, most of the prevalent CNN-based models perform inference using static convolution kernels, which may limit their representation power, efficiency, and ability for interpretation. To cope with this challenge, dynamic convolution (Chen et al., 2020) is proposed by aggregating parallel convolution kernels via attention mechanism (Vaswani et al., 2017). Compared to static models, which have fixed computational graphs and parameters at the inference stage, dynamic networks can adapt their structures or parameters to different inputs, leading to notable advantages in terms of accuracy, computational efficiency, adaptiveness, etc (Han et al., 2021). However, it is challenging to jointly optimize the attention score and the static kernels in dynamic convolution. To mitigate the joint optimization difficulty, Li et al. (2021b) revisited it from the matrix decomposition perspective by reducing the dimension of the latent space.

In this work, we pioneer a novel deep learning-based solution, titled DynaPicker, for seismic body wave phase classification. Furthermore, the phase classifier trained on the short-window data is used to estimate the arrival times of the P-wave and S-wave on the continuous waveform on a long-time scale. In DynaPicker, the 1D dynamic convolution decomposition (DCD) adapted from the work of Li et al. (2021b) on image classification is used as the backbone of the solution (see Figure 1 for illustration).

In order to complete seismic body wave phase classification and phase onset time picking, the main steps in this study are included as follows. First, the impact of different input data lengths on the performance of seismic phase detection and arrival time picking are studied on the subset of the STanford EArthquake Dataset (STEAD) (Mousavi et al., 2019a). Then, the SCEDC dataset (Center, 2013) without specific phase arrival-time labeling collected by the Southern California Seismic Network is used to train and test the model in seismic phase identification. Finally, the pre-trained model is further applied to several open-source seismic datasets to evaluate the model performance in phase arrival-time picking performance. To that aim, in this study, the STEAD dataset(Mousavi et al., 2019a), the Italian seismic dataset for machine learning (INSTANCE) (Michelini et al., 2021), and the dataset across the Iquique region of northern Chile (Iquique) (Woollam et al., 2019) are used to verify the model performance in seismic phase picking.

The main contributions of this case study are summarized as follows:

- – This case study first introduces a deep learning-based seismic phase identification solution, called DynaPicker, which is capable of reliably detecting P- and S-waves of even very small earthquakes, e.g., the local magnitude of the SCEDC dataset ranges from $-0.81$ to $5.7$ $M_L$.

- – The results tested on the data of varying lengths indicate that DynaPicker can be adaptive to different lengths of input data for seismic phase identification. Meanwhile, it is proved that Dynapicker is robust to classify seismic phases even when the seismic data is polluted by noise.

- – The testing data and the training data used for seismic phase identification and phase picking have no overlap, which proves that DynaPicker is capable of generalizing entire waveforms and metadata archives from different regions.

- – The CREIME model (Chakraborty et al., 2022b) is used to perform magnitude estimation for waveform windows for which the P-wave probability surpasses the threshold of 0.7. The results are highly dependent on this threshold, and it should be chosen after carefully looking at the data. It might also be necessary to use different thresholds for different stations.

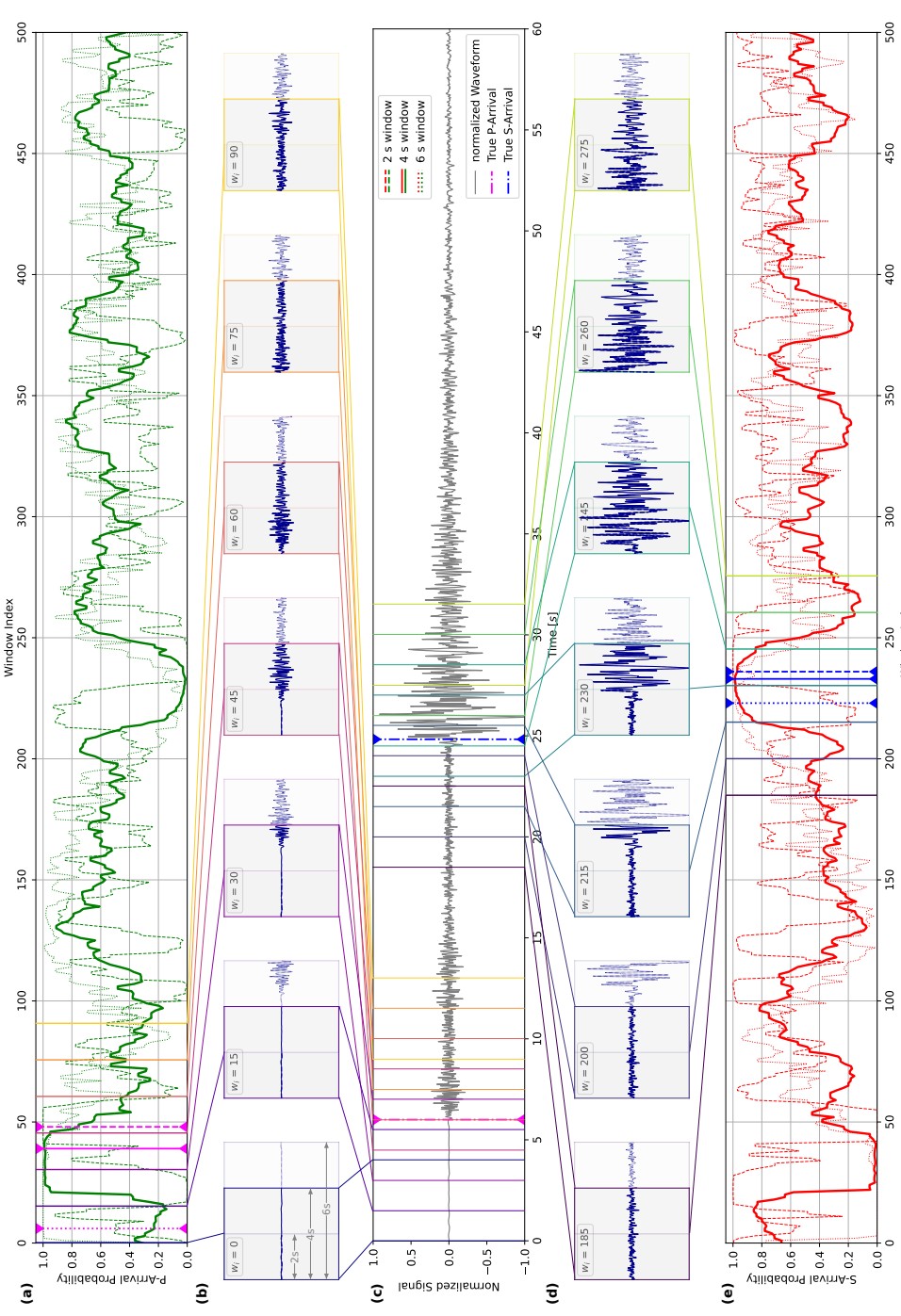

**Figure 2.** Visualization of arrival-time picking using DynaPicker for a given normalized seismic waveform. Here, the subfigure **c** shows only one channel of a real seismogram from the STEAD dataset (Mousavi et al., 2019a). The figure presents the model performance for different input window lengths of 2s, 4s, and 6s; the windows are shifted by 10 samples at a time (for further details on this refer to the methodology section). The subsequent windows are denoted by different colors and shown explicitly in subfigures **b** and **d**. Note that we only show specific windows around P- and S-arrivals in subfigures **b** and **d**, respectively, as they are most relevant for the corresponding picks. **a** and **e** show the predicted probability of P-phase and S-phase arrivals, respectively for the entire waveform. Each window visualized in subfigure **b**, is mapped to a vertical line of the corresponding color in subfigure **a** at the window index $w_i$ representing that window. Similarly, each window visualized in subfigure **d**, is mapped to a vertical line of the corresponding color in subfigure **e** at the window index representing that window. The blue and pink dashed vertical lines in subfigure **c** represent the true P-phase and S-phase arrival times (provided in the metadata for the dataset), respectively; analogously, the solid dashed and dotted blue vertical lines in subfigure **a** indicate the window indices corresponding to the predicted P-arrival for models trained on 4s, 2s and 6s windows respectively and the solid, dashed and dotted pink vertical lines in subfigure **e** indicate the window indices corresponding to the predicted S-arrival for models trained on 4s, 2s and 6s windows respectively. The P- and S-arrival samples are considered to be at the center of the picked windows.

## 2 Methodology

In this study, we develop a 1D-DCD-based seismic phase classifier to handle seismic time series data. Our model takes a window of the normalized three-channel seismic waveform as input and predicts its label as P-phase, S-phase, or noise. Then, the pre-trained model is employed to automatically pick the arrival time on the continuous seismic data. Figure 1 schematically visualizes the proposed model architecture which consists of convolutional layers, batch normalization, dropout, DCD-based layers, and a 1D dynamic classifier adapted from the work (Li et al., 2021b).

### 2.1 Dynamic convolution decomposition (DCD)

Dynamic convolution achieves a significant performance improvement over convolutional neural networks (CNNs) by adaptively aggregating $K$ static convolution kernels (Yang et al., 2019; Chen et al., 2020). As shown in the paper written by Li et al. (2021b), based on an input-dependent attention mechanism, dynamic convolution succeeds in aggregating multiple convolution kernels into a convolution weight matrix, which can be described as Eq. (1) and (2).

$$\boldsymbol{W}(\boldsymbol{x}) = \sum_{k=1}^{K} \pi_k(\boldsymbol{x})\boldsymbol{W}_k \tag{1}$$

$$s.t. \quad 0 \leq \pi_k(\boldsymbol{x}) \leq 1, \sum_{k=1}^{K} \pi_k(\boldsymbol{x}) = 1 \tag{2}$$

where the attention scores $\{\pi_k(\boldsymbol{x})\}$ are used to linearly aggregate the $K$ convolution kernels $\{\boldsymbol{W}_k(x)\}$.

However, the vanilla dynamic convolution suffers from two main limitations: firstly, the use of $K$ kernels will lead to the lack of compactness; secondly, it is challenging to jointly optimize the attention scores $\{\pi_k(\boldsymbol{x})\}$ and static kernels $\{\boldsymbol{W}_k\}$ (Li et al., 2021b).

To address the above-mentioned issues, Li et al. (2021b) revisited dynamic convolution from a matrix decomposition viewpoint. They further proposed dynamic channel fusion to replace dynamic attention over channel group to reduce the dimension of the latent space, and mitigate the difficulty of the joint optimization problem. Figure 1 gives an illustration of a DCD layer. The general formulation of dynamic convolution using dynamic channel fusion is given (Li et al., 2021b):

$$\boldsymbol{W}(\boldsymbol{x}) = \boldsymbol{\Lambda}(\boldsymbol{x})\boldsymbol{W}_0 + \boldsymbol{P}\boldsymbol{\Phi}(\boldsymbol{x})\boldsymbol{Q}^T \tag{3}$$

where $\boldsymbol{\Lambda}(\boldsymbol{x})$ represents a $C \times C$ diagonal matrix ($C$ denotes the number of channels), and $\boldsymbol{W}_0$ denotes the static kernel . In the matrix $\boldsymbol{\Lambda}(\boldsymbol{x})$, the element $\lambda_{i,i}(\boldsymbol{x})$ is a function of the input $\boldsymbol{x}$. The matrix $\boldsymbol{\Phi}(\boldsymbol{x})$ of size $L \times L$ fuses channels in the latent space $\mathbb{R}^L$ associated with the dimensionality $L$ dynamically. The two static matrices $\boldsymbol{Q} \in \mathbb{R}^{C \times L}$ and $\boldsymbol{P} \in \mathbb{R}^{L \times L}$ are used to compress the input $\boldsymbol{x}$ into a low dimensional space and expand the channel number to the output space, respectively (More details can be found in the paper (Li et al., 2021b)).

## 2.2 Seismic phase classifier network architecture

As presented in Figure 1, the first convolutional layer is applied to process a three-channel window of seismic data in the time domain, to generate a feature representation. Then, a batch normalization layer (BN) is used to accelerate the training process and provide stability for the network followed by an activation function using Rectified Linear Unit (ReLU) (Agarap, 2018). Finally, a max-pooling block (Simonyan and Zisserman, 2015) is added to reduce the size of the feature map, which is followed by a Dropout layer (Srivastava et al., 2014) to avoid overfitting. The second part of the framework is comprised of several DCD-based layers, which are used to leverage favorable properties that are absent in static models. The right part of Figure 1 shows the diagram of the 1D-DCD block, where a dynamic branch is used to produce coefficients for dynamic channel-wise attention $\mathbf{\Lambda}(\boldsymbol{x})$ of size $C \times C$ and dynamic channel fusion $\mathbf{\Phi}(\boldsymbol{x})$ of size $L \times L$ (Li et al., 2021b). In the dynamic branch, the average pooling is first applied to the input $\boldsymbol{x}$ and then is followed by two fully connected (FC) layers associated with an activation layer between them. For the two used FC layers, the former aims to reduce the number of channels, and the latter tries to expand them into $C + L^2$ outputs. Similar to a static convolution, a DCD layer also includes a batch normalization and an activation (e.g. ReLU) layer followed by a dropout layer. Finally, the dynamic classifier uses this information to map the high-level features to a discrete probability over 3 categories (P-wave, S-wave, and noise wave). The dynamic classifier is also based on a 1D-DCD block.

It is worth noting that the model introduced in this study can be easily adapted to address inputs with different window sizes by simultaneously adjusting the sizes of the first layer and the dynamic classifier layer, respectively. With the goal to verify the model robustness, the impact of different length data on seismic phase identification is investigated in the following section. The pre-trained model is extensively applied to pick arrival time on continuous data. The process of the arrival time picking using different window sizes when feeding the same continuous seismic waveform is schematically visualized in Figure 2.

## 2.3 Phase arrival-time estimation

To achieve seismic phase picking, the following steps are included, where the main steps are the same as in GPD (Ross et al., 2018) and CapsPhase (Saad and Chen, 2021). The pipeline for phase arrival time picking on continuous seismic data using the pre-trained phase classifier is visualized in Figure 3.

 – First, each waveform is filtered using the bandpass filter. For instance, the data from the STEAD dataset is filtered within the frequency range 2-20 Hz, following the CapsPhase (Saad and Chen, 2021).

 – Then, the waveform is sampled at 100 Hz followed by normalization using the absolute maximum amplitude. For example, for the STEAD dataset, each waveform has a size of $6000 \times 3$ after pre-processing.

 – Afterwards, the data of filtered are divided into several windows. Each window contains a 4s-three-component seismogram (400 samples since the sampling rate is 100 Hz), while the window strides with ten samples such that the number of overlapping samples between neighbor windows is 390 samples. Therefore, the total number of windows is as follows.

$$N_{win} = \frac{L_{total} - L_{win}}{n_{shift}} + 1 \tag{4}$$

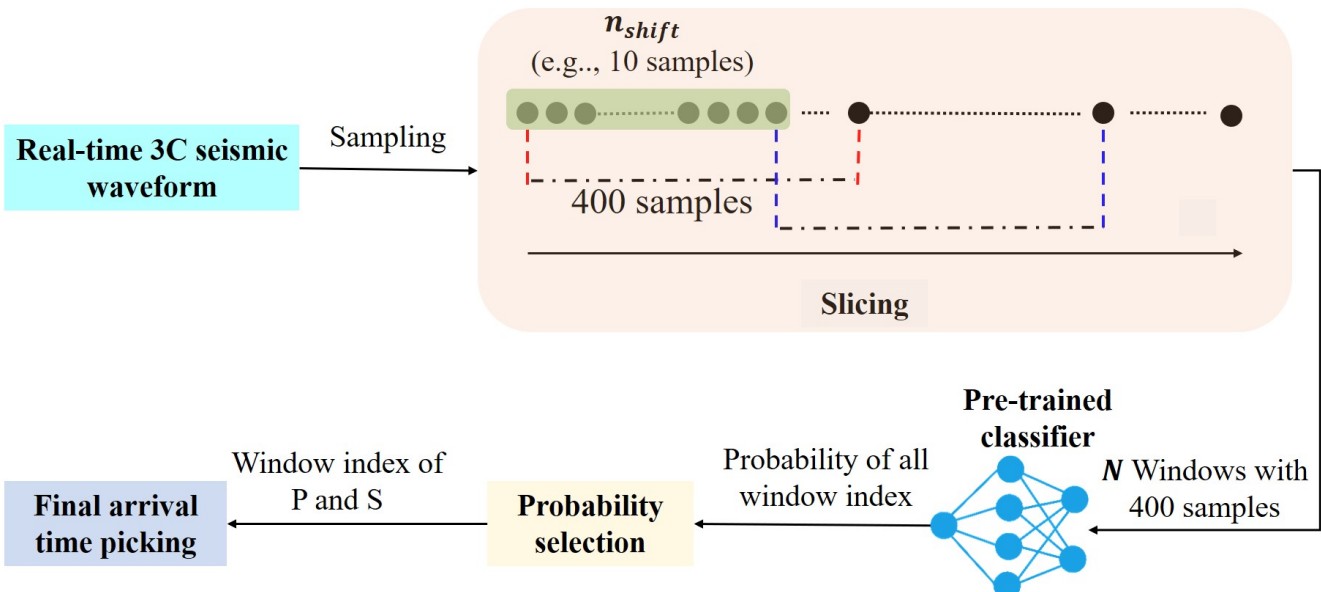

**Figure 3.** Pipeline of arrival time picking for continuous seismic waveforms given the pre-trained classifier.

where $L_{total}$ and $L_{win}$ denote the length of the original waveform after sampling, and the length of the window (e.g., 400 samples in this study), respectively. $n_{shift}$ is the number of the shift between windows in samples and in this work, it is empirically set as 10 same as CapsPhase (Saad and Chen, 2021).

– Then, the pre-trained classifier is utilized to predict three sequences of probabilities for each window associated with P-phase, S-phase, and noise respectively. Following the work (Chen et al., 2020), a temperature softmax function (Goodfellow et al., 2016) is used in this study to smooth the output probability as follows:

$$\delta_k = \frac{\exp\left(z_k/T\right)}{\sum_j \exp\left(z_j/T\right)} \tag{5}$$

where $z_k$ is the output of the classifier layer, and $T$ is the temperature. The original softmax function is a special case when $T = 1$. As $T$ increases, the output is less sparse. In this study, the value of $T$ is experimentally set to 4 [1].

– Finally, the arrival-time detection is declared using the following equation:

$$t_{(P/S)} = 0.01 \times (Win_{index} \times n_{shift} + n_c) + t_{start} \tag{6}$$

where $Win_{index}$ denotes the window index of the largest probability, and $t_{start}$ is the trace starting time. $n_c$ denotes the added constant that is $0.5 \times$ window length for generalization since in the SCEDC dataset (Center, 2013), the P-wave and S-wave windows are centered around the arrival time.

[1] It is important to mention that the pre-trained versions of GPD and CapsPhase have the softmax function's $T$ value, used for seismic phase classification, set to 1. Therefore, in this study, the temperature $T$ is specifically set to 4 solely for the phase picking process.

## 3 Data and labeling

In this work, the dataset provided by Southern California Earthquake Data Center (SCEDC) (Center, 2013) is used for model training and testing in seismic phase identification. The magnitude range of the data is $-0.81 < M_L < 5.7$. This dataset is comprised of 4.5 million three-component seismic signals with a duration of 4s including 1.5 million P-phase picks, 1.5 million S-phase picks, and 1.5 million noise windows. The P-wave and S-wave windows are centered on the arrival pick, while each noise window is captured by starting 5s before each P-wave arrival. Finally, the absolute maximum amplitude discovered on the three components is used to normalize each three-component seismic record. In this study, 90% of the seismograms from the SCEDC dataset (Center, 2013) are used for model training, and the remaining 5% of seismograms are employed to test the model performance. Furthermore, we compare the seismic phase classification performance to a capsule neural network-based seismic data classification approach, termed CapsPhase (Saad and Chen, 2021), and our previous work, 1D-ResNet (Li et al., 2022b).

To achieve seismic phase identification, DynaPicker takes a window of three-channel waveform seismogram data as input, and then for each input, the model predicts the probabilities corresponding to each class (P-wave, S-wave, or noise). This model has three output labels: zero for the P-wave window, one for the S-wave window, and two for the noise window.

In order to further evaluate the model performance in phase arrival-time picking pre-trained on the SCEDC dataset (Center, 2013), several subsets of three open-source public seismic datasets namely the STEAD dataset (Mousavi et al., 2019a), the INSTANCE dataset (Michelini et al., 2021), and the Iquique dataset (Woollam et al., 2019) are used. Each waveform in the first two datasets is either 1 or 2 minutes long. They can be viewed as good generalization tests of our proposed method. DynaPicker is compared to the generalized phase detection (GPD) framework (Ross et al., 2018) based on convolutional neural networks, CapsPhase (Saad and Chen, 2021) based on capsule neural network (Sabour et al., 2017), and AR picker (Akazawa, 2004) to evaluate the performance of phase arrival-time picking on continuous seismic recordings.

## 4 Evaluation metrics for seismic phase classification

In this article, noise labels are not treated differently from phase labels, so classifying a noise window correctly has the same weight as confirming a phase window. The seismic phase detector can be viewed as a three-class classifier that decides whether a given time window contains a seismic phase (P or S), or only noise. Here, the "noise" windows do not contain P- or S-phases. We can evaluate a deep-learning model by processing labeled testing data where the true output is known. The accuracy defined below is a simple measure of a classifier's performance:

$$Accuracy = \frac{N_C}{N_T} \tag{7}$$

where $N_C$ denotes the number of correctly labeled samples and $N_T$ represents the total number of testing samples.

To evaluate the detector's effectiveness, a confusion matrix (Stehman, 1997) is adopted to reflect the classification result, and then precision and recall can be defined as follows:

$$Precision = \frac{TP}{TP + FP} \tag{8}$$

$$Recall = \frac{TP}{TP + FN} \tag{9}$$

The F1-score is computed from the harmonic mean of precision and recall for each class:

$$F1_{score} = 2 \times \frac{Precision \times Recall}{Precision + Recall} \tag{10}$$

where TN, FN, FP, and TP are the number of true negative, false negative, false positive, and true positive, respectively.

## 5 Experiments and Results

### 5.1 Seismic phase classifier training

In this study, for dynamic convolution decomposition units, all the weight and filter matrices are initialized with a normal initializer and bias vectors set to zeros. For optimization, we use the ADAM (Kingma and Ba, 2014) algorithm, which keeps track of first- and second-order moments of the gradients and was invariant to any diagonal rescaling of the gradients. We used a learning rate of $10^{-3}$ and trained the DynaPicker for 50 epochs same as CapsPhase (Saad and Chen, 2021). In this work, DynaPicker was implemented in PyTorch (Paszke et al., 2019), and all the training was performed on an NVIDIA A100 GPU. The model was trained using a cross-entropy loss function with the ADAM optimization algorithm, in mini-batches of 480 records. We used a dropout rate of 0.2 for all dropout layers.

### 5.2 Investigation on different length input data

Here, we investigate the impact of different input data lengths on the performance of seismic phase detection and arrival-time picking using the STEAD dataset. The details of arrival-time picking using a pre-trained phase classifier can be found in the following subsections and the Methods section.

#### 5.2.1 Different length of the input data on phase classification

To that end, we select 58,018 earthquake waveforms from the STEAD dataset (Mousavi et al., 2019a) and create three datasets within different durations (2s, 4s, and 6s). There is a total of 174,054 waveforms including P-wave, S-wave, and noise wave in each dataset. In this experiment, all data are re-sampled at 100 Hz and each three-component waveform is normalized by the absolute maximum amplitude observed on any of the three components. Similar to the SCEDC dataset (Center, 2013), P-wave and S-wave windows are centered on the respective arrival-time picks. Meanwhile, noise windows are captured from pure noise waveforms. Note that these three datasets are comprised by the same events, and only the window length is different.

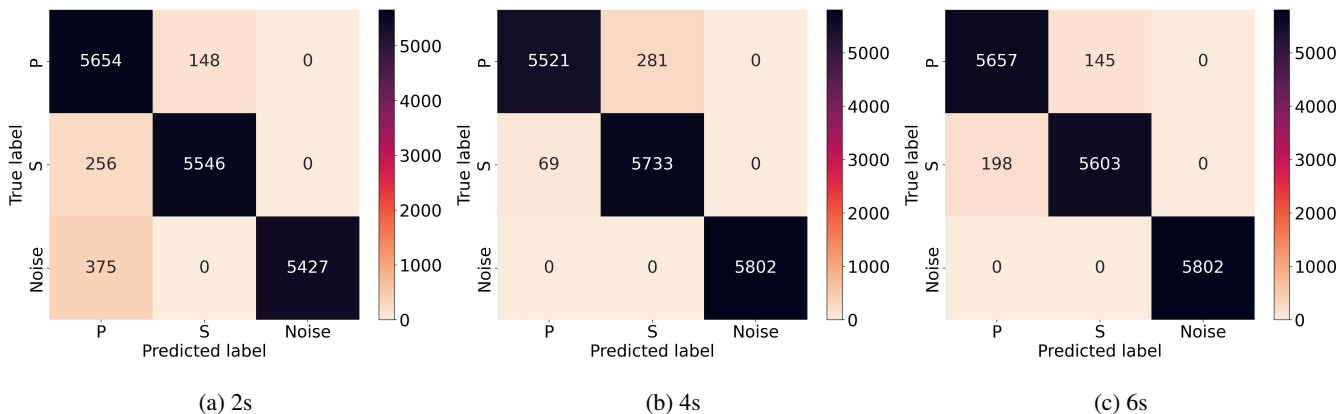

**Figure 4.** Confusion matrices for seismic phase classification given different length input data: (a) 2s, (b) 4s, and (c) 6s using DynaPicker.

**Table 1.** Body-wave arrival time evaluation using different window lengths on the STEAD dataset.

| Method | No. of undetected events | No. of abs(e) $\leq$ 0.5s for P-pick | $\mu_P$ | $\sigma_P$ | No. of abs(e) $\leq$ 0.5s for S-pick | $\mu_S$ | $\sigma_S$ |
|---|---|---|---|---|---|---|---|
| DynaPicker (2s) | 0 | 8236 | 0.027 | 0.146 | 3014 | -0.048 | 0.200 |
| DynaPicker (4s) | 0 | 3734 | 0.011 | 0.136 | 3855 | -0.120 | 0.182 |
| DynaPicker (6s) | 0 | 2819 | 0.058 | 0.218 | 1733 | -0.127 | 0.224 |
| EPick | 0 | 9873 | -0.002 | 0.052 | 9663 | 0.002 | 0.122 |

$\mu_P$ and $\sigma_P$ are the mean and standard deviation of errors (ground truth $-$ prediction) in seconds respectively for P phase picking.

$\mu_S$ and $\sigma_S$ are the mean and standard deviation of errors (ground truth $-$ prediction) in seconds respectively for S phase picking.

Then, each dataset is split into a training dataset (90%) and a testing dataset (10%). The overall testing accuracy for different length input data is estimated to be 95.52%, 97.99%, and 98.02% in line with 2s, 4s, and 6s respectively. The result demonstrates that DynaPicker can work well with the input of different time duration.

The confusion matrices corresponding to the input data with different duration are shown in Figure 4. We can observe that the developed model reaches a high detection accuracy for each class, especially in noise window detection as shown in Figure 4, where noise waveform is more easily distinguishable from P and S arrivals than they are from each other in the cases for 4s and 6s data.

In the end, the testing results indicate that our model can be adaptive to different lengths of input data. At the same time, our model achieves a compatible performance in seismic phase picking even with low-volume training data.

### 5.2.2 The impact of different lengths of the input data for continuous seismic record

In this part, the pre-trained DynaPicker on the seismic data with different time duration are further evaluated on continuous seismic data. Meanwhile, the model is compared to EPick (Li et al., 2022a), a simple neural network that incorporates an

**Table 2.** Results of evaluation metrics on the test dataset Center (2013) for phase classification.

| Category | Model | Precision | Recall | F1-score | Ref. |
|----------|-------|-----------|--------|----------|------|
| P-phase | DynaPicker | **99.15%** | 98.54% | **98.84%** | This study |
|         | CapsPhase | 98.93% | 98.45% | 98.69% | Saad and Chen (2021) |
|         | 1D-ResNet | 98.88% | **98.64%** | 98.76% | Li et al. (2022b) |
| S-phase | DynaPicker | 98.87% | **99.04%** | **98.96%** | This study |
|         | CapsPhase | **98.89%** | 98.63% | 98.76% | Saad and Chen (2021) |
|         | 1D-ResNet | 98.72% | 98.94% | 98.83% | Li et al. (2022b) |
| Noise | DynaPicker | 98.43% | 98.86% | **98.65%** | This study |
|       | CapsPhase | 98.17% | **98.90%** | 98.54% | Saad and Chen (2021) |
|       | 1D-ResNet | **98.52%** | 98.54% | 98.53% | Li et al. (2022b) |

The saved model of CapsPhase is directly used here without retraining and, unlike the original CapsPhase(Saad and Chen, 2021), the output threshold for each class is not used in this work since it reduces the CapsPhase performance in the testing phase. Bold values represent the best performance.

attention mechanism into a U-shaped neural network. Here, the pre-trained and saved model of EPick is directly used without retraining. Besides, there is no overlap between the training data used for seismic phase identification and the data adopted in testing the model performance in phase picking. The testing results are summarized in Table 1. Here, we can observe that EPick achieves the best performance in phase picking over DynaPicker by using different window sizes. The potential reason is that EPick is pre-trained on the data labeled with the specific phase arrival time from the STEAD dataset. Secondly, a larger window size reduces the amount of the P-phase with an error lower than 0.5s. Thirdly, in the case where the window size is 4s, the number of the S-phase with an error lower than 0.5s is larger than in the other two cases e.g., 2s and 6s.

## 5.3  Seismic phase classification on 4s SCEDC Dataset

As discussed in the previous subsections, the proposed model, DynaPicker, can be adapted to the data with different lengths and achieves compatible performance.

Here, DynaPicker is further retrained and tested on the SCEDC dataset (Center, 2013) collected by the Southern California Seismic Network (SCSN) [2]. Then, we compared our model with CapsPhase (Saad and Chen, 2021) and our previous work, 1D-ResNet (Li et al., 2022b) with the same test set. The testing accuracy of DynaPicker is **98.82**%, which is slightly greater than CapsPhase (Saad and Chen, 2021) (98.66%) and 1D-ResNet (Li et al., 2022b) (98.66%).

Then, different evaluation metrics, like the Precision, Recall, and F1-score for DynaPicker, CapsPhase (Saad and Chen, 2021), and 1D-ResNet (Li et al., 2022b) are summarized in Table 2. As one can see from Table 2, compared with the baseline methods, DynaPicker can achieve superior performance in terms of the F1-score. For precision and recall, DynaPicker also achieves a comparable performance.

---

[2]For further information regarding the retraining of CapsPhase and GPD, and the alterations made to DynaPicker, please refer to the Discussion section, where you can find comprehensive details on these topics.

**Table 3.** Testing results of different noise levels for phase identification on the STEAD dataset.

| Noise level | 0.01 | 0.05 | 0.1 | 0.15 |
|---|---|---|---|---|
| Capsphase (Saad and Chen, 2021) | 95.28% | 95.43% | 92.80% | 88.90% |
| 1D-ResNet (Li et al., 2022b) | 96.30% | 96.46% | 93.22% | 89.16% |
| DynaPicker | **96.88%** | **96.73%** | **94.49%** | **91.26%** |

The best-saved model of CapsPhase is directly used here without retraining and unlike the original
CapsPhase paper (Saad and Chen, 2021), the output threshold for each class is not used in this work since it
reduces the CapsPhase performance in the testing phase. Bold values represent the best performance.

Finally, in order to investigate the model performance, when facing more noisy data, the same subset selected from the STEAD dataset used in 1D-ResNet (Li et al., 2022b) is utilized. Here, the signal-to-noise ratio (SNR) of the selected data before adding noise ranges from 0 to 70 dB, and the SNR is the mean value of SNR over three components for each signal. The magnitude of the data ranges from 1.0 to 3.0. To study the impact of different noise levels on model performance, the subset is masked by the Gaussian noise (similar to the method used in EQTransformer (Mousavi et al., 2020)) with mean $\mu = 0$ and standard deviation $\delta = 0.01, 0.05, 0.1,$ and $0.15$, respectively. Afterward, these noisy data are fed to the pre-trained phase classifier to test the model performance. The testing accuracies of different models are summarized in Table 3 below. The results in Table 3 show that (a) large noise reduces the model performance; (b) DynaPicker outperforms over CapsPhase and 1D-ResNet; (c) DynaPicker is robust in identifying seismic phases when the seismic data is polluted by noise.

## 5.4 Seismic arrival-time picking on continuous seismic records

We next demonstrate the applicability of our model to pick the seismic phase arrival time for continuous seismic data in the time domain. The main parameters related to phase arrival-time picking are studied in the following section. Within this work, DynaPicker is implemented for seismic phase identification given short-window seismic waveforms same as GPD and CapsPhase. Hence, DynaPicker is firstly compared with two window-based methods including GPD and CapsPhase on both the STEAD dataset and the INSTANCE dataset. Secondly, we compare DynaPicker with one of the state-of-the-art sample-based seismic phase pickers, EQTransformer (Mousavi et al., 2020) on the Iquique dataset (Woollam et al., 2019). The reason is that on one hand, EQTransformer is a multi-task deep learning model designed for earthquake detection and seismic phase picking, which is trained on the STEAD dataset labeled with specific phase arrival time. On the other hand, the original INSTANCE paper (Michelini et al., 2021) reported that EQTransformer is used in picking the first arrivals of P- and S- waves. Therefore, in this study, the subset of the Iquique dataset (Woollam et al., 2019) is further applied to achieve a fair comparison between DynaPicker and EQTransformer.

### 5.4.1 Comparison with window-based methods

• **Application to the STEAD dataset.** We randomly select $2 \times 10^4$ earthquake waveforms from the STEAD dataset out of which $1 \times 10^4$ have a time difference greater than 4s between P-wave arrival and S-wave arrival, while for the remainder

**Table 4.** Body-wave arrival time evaluation using different methods on STEAD dataset including (a) $S_{arrival} - P_{arrival} > 4s$ and (b) $S_{arrival} - P_{arrival} < 4s$. In each case, $1 \times 10^4$ samples are used. Following Saad and Chen (2021), the event whose pick predicted by a model has an absolute error larger than 0.5 s, is recognized as false positive.

(a) $S_{arrival} - P_{arrival} > 4s$

| Method | No. of events detected | No. of abs(e) $\leq$ 0.5s for P-pick | $\mu_P$ | $\sigma_P$ | No. of abs(e) > 0.5s for P-pick | No. of abs(e) $\leq$ 0.5s for S-pick | $\mu_S$ | $\sigma_S$ | No. of abs(e) > 0.5s for S-pick |
|---|---|---|---|---|---|---|---|---|---|
| DynaPicker | 10000 | **9055** | **0.0002** | 0.151 | 945 | **7696** | **0.011** | 0.203 | 2304 |
| GPD | 9826 | 8975 | -0.0036 | 0.149 | 851 | 2623 | -0.043 | 0.193 | 7203 |
| CapsPhase | 9885 | 8766 | -0.018 | 0.149 | 1119 | 5545 | -0.112 | 0.184 | 4340 |
| AR picker | 10000 | 7963 | 0.079 | 0.133 | 2037 | 4011 | 0.205 | 0.176 | 5989 |

(b) $S_{arrival} - P_{arrival} < 4s$

| Method | No. of events detected | No. of abs(e) $\leq$ 0.5s for P-pick | $\mu_P$ | $\sigma_P$ | No. of abs(e) > 0.5s for P-pick | No. of abs(e) $\leq$ 0.5s for S-pick | $\mu_S$ | $\sigma_S$ | No. of abs(e) > 0.5s for S-pick |
|---|---|---|---|---|---|---|---|---|---|
| DynaPicker | 10000 | **9405** | **0.0048** | 0.091 | 595 | **7597** | **0.0075** | 0.179 | 2403 |
| GPD | 9662 | 8890 | 0.0059 | 0.092 | 772 | 4393 | -0.012 | 0.164 | 5269 |
| CapsPhase | 9861 | 8767 | -0.020 | 0.084 | 1094 | 5545 | -0.061 | 0.164 | 4316 |
| AR picker | 10000 | 7755 | 0.015 | 0.075 | 2245 | 7369 | 0.126 | 0.161 | 2361 |

$\mu_P$ and $\sigma_P$ are the mean and standard deviation of errors (ground truth − prediction) in seconds respectively for P phase picking. $\mu_S$ and $\sigma_S$ are the mean and standard deviation of errors (ground truth − prediction) in seconds respectively for S phase picking.

this difference is lower than 4s and the epicentral distances are less than or equal to 35 km. Here we use 4s as the threshold for waveform selection since the SCEDC dataset (Center, 2013) with the duration of 4s is used to train and test the model performance on seismic phase classification. To study the impact of the time difference between P and S picks, the events of different time differences are used to verify our model's robustness in seismic arrival time picking for continuous seismic data.

As presented in GPD (Ross et al., 2018) and CapsPhase (Saad and Chen, 2021), a triggering method is used to locate arrival picks by setting a threshold. However, the picking performance is impacted by the threshold. To overcome this drawback we use the window index with the largest probability to locate the P and S picks as this empirically yields the best results.

Table 4 summarizes the testing result of arrival-time picking on the STEAD dataset. From Table 4, we can see that (a) our model succeeds in correctly detecting all seismic events, while about 174 and 115 seismic events cannot be detected by GPD (Ross et al., 2018) and CapsPhase (Saad and Chen, 2021) for the earthquake signal with $S_{arrival} - P_{arrival} > 4s$, and about 338 and 139 seismic events cannot be detected by GPD (Ross et al., 2018) and CapsPhase (Saad and Chen, 2021) for the earthquake signal with $S_{arrival} - P_{arrival} < 4s$; (b) compared with GPD (Ross et al., 2018), CapsPhase (Saad and Chen, 2021), and AR picker (Akazawa, 2004), most of the error between the located P-wave or S-wave picks and the ground truth are

**Table 5.** Body-wave arrival time evaluation using different methods on INSTANCE dataset including (a) $S_{arrival} - P_{arrival} > 4s$ and (b) $S_{arrival} - P_{arrival} < 4s$. In each case, $1 \times 10^4$ samples are used. Following Saad and Chen (2021), the event whose pick predicted by a model has an absolute error larger than 0.5 s, is recognized as false positive.

(a) $S_{arrival} - P_{arrival} > 4s$

| Method | No. of events detected | No. of abs(e) $\leq 0.5s$ for P-pick | $\mu_P$ | $\sigma_P$ | No. of abs(e) $> 0.5s$ for P-pick | No. of abs(e) $\leq 0.5s$ for S-pick | $\mu_S$ | $\sigma_S$ | No. of abs(e) $> 0.5s$ for S-pick |
|---|---|---|---|---|---|---|---|---|---|
| DynaPicker | 10000 | **8707** | 0.030 | 0.130 | 1293 | **7530** | **0.019** | 0.199 | 2470 |
| GPD | 9623 | 8231 | 0.028 | 0.123 | 1392 | 4726 | -0.032 | 0.179 | 4897 |
| CapsPhase | 9598 | 7948 | 0.014 | 0.140 | 1650 | 5837 | -0.103 | 0.186 | 3761 |
| AR picker | 9999 | 7545 | 0.052 | 0.118 | 2454 | 3274 | 0.218 | 0.168 | 6725 |

(b) $S_{arrival} - P_{arrival} < 4s$

| Method | No. of events detected | No. of abs(e) $\leq 0.5s$ for P-pick | $\mu_P$ | $\sigma_P$ | No. of abs(e) $> 0.5s$ for P-pick | No. of abs(e) $\leq 0.5s$ for S-pick | $\mu_S$ | $\sigma_S$ | No. of abs(e) $> 0.5s$ for S-pick |
|---|---|---|---|---|---|---|---|---|---|
| DynaPicker | 10000 | **8690** | **0.012** | 0.079 | 1310 | **7815** | **0.0085** | 0.160 | 2185 |
| GPD | 9833 | 8109 | 0.022 | 0.075 | 1724 | 6647 | -0.019 | 0.134 | 3186 |
| CapsPhase | 9872 | 7984 | 0.019 | 0.091 | 1888 | 5447 | -0.072 | 0.143 | 4325 |
| AR picker | 10000 | 8296 | 0.016 | 0.077 | 1704 | 5778 | 0.149 | 0.168 | 4222 |

$\mu_P$ and $\sigma_P$ are the mean and standard deviation of errors (ground truth − prediction) in seconds respectively for P phase picking. $\mu_S$ and $\sigma_S$ are the mean and standard deviation of errors (ground truth − prediction) in seconds respectively for S phase picking.

within 0.5 s when using DynaPicker. We use 0.5 s for our analysis following CapsPhase(Saad and Chen, 2021); (c) DynaPicker is robust for seismic events of different time differences between P and S picks. In summary, our proposed model outperforms the baseline methods.

• **Application to the INSTANCE dataset.** We also evaluate the picking performance of our model using the INSTANCE dataset (Michelini et al., 2021) and compare the picking error with the benchmark methods. This dataset includes about 1.2 million three-component waveforms from about $5 \times 10^4$ earthquake events recorded by the Italian National Seismic Network. In the metadata, the recorded earthquake list ranges from January 2005 to January 2020, and the magnitude of the earthquake events ranges from 0.0 to 6.5. All the recorded seismic waveforms have a duration of 120 s and are sampled at 100 Hz. We randomly select $2 \times 10^4$ earthquake waveforms from the INSTANCE dataset (Michelini et al., 2021), out of which $1 \times 10^4$ have a time difference greater than 4 s between P-wave arrival and S-wave arrival, while for the remainder this difference is lower than 4s and similarly, the epicentral distances are less than or equal to 35 km.

As summarized in Table 5, we can observe that the proposed model outperforms the baseline methods. On one hand, the proposed model succeeds in identifying the true label corresponding to each input, which means all seismic events are detected

compared with the used baseline methods. On the other hand, our model achieves a lower arrival-time picking error, and it is robust for different time differences between P and S picks. In particular, our model achieves the lowest mean error in S-phase arrival-time picking for both cases.

### 5.4.2 Comparison with sample-based method

The Iquique dataset is comprised of locally recorded seismic arrivals throughout northern Chile and is used in several previous studies (Woollam et al., 2019, 2022; Münchmeyer et al., 2022) to train a deep learning-based picker. It contains about $1.1 \times 10^4$ manually picked P-/S- phase pairs, where all the seismic waveform units are recorded in counts. In this study, $1 \times 10^4$ P-/S-phase pairs are used, and DynaPicker is further compared with the advanced deep learning model Earthquake transformer (EQTransformer) (Mousavi et al., 2020), to evaluate onset picking. In particular, it is worth noting that neither DynaPicker nor EQTransformer is retrained on the Iquique dataset.

First, the confusion matrices for P- and S- arrival picking results of the experiment using DynaPicker and EQTransformer are shown in Figure 5a and 5b. We find that out of the selected $1 \times 10^4$ signals, EQTransformer misses 243 events, which means that for these misclassified earthquake events, no arrival pick is detected. Compared with EQTransformer, DynaPicker is capable of detecting all earthquake events including all P-phase and S-phase arrival time pairs.

Then, two examples from the Iquique dataset using EQTransformer and DynaPicker are displayed in Figure 5c and 5d, respectively. The picking result of EQTransformer is implemented by using Seisbench (Woollam et al., 2022), and in DynaPicker, only the sample of the largest probability is recognized as P- or S-phases. It can be observed that in Figure 5c the detected P-phase by EQTransformer is with a low probability, and the S-phase is missing, while the estimated P-phase by DynaPicker is of high probability, and S-phase is also detected as shown in the bottom subplot of Figure 5c. In Figure 5d, EQTransformer detects multiple picks including one incorrectly detected P-phase, and DynaPicker also picks multiple P-phase and S-phases. In contrast to EQTransformer, in DynaPicker only the sample with the largest probability is regarded as the true prediction for both P- and S-phases. However, as shown in the bottom subplot of Figure 5d, DynaPicker is capable of determining the truly predicted P- and S-phases with a larger probability compared to EQTransformer.

Finally, the absolute error between deep learning-based model predicted picks (e.g., EQTransformer and DynaPicker) and manual picks that are below 0.5s are taken into account. For both P and S waves, EQTransformer performs slightly better than DynaPicker in terms of both the root mean squared error (RMSE) and the mean absolute error (MAE). Here, the MAE and RMSE of both P and S waves using EQTransformer are $MAE(P) = 0.091s$, $RMSE(P) = 0.095s$ and $MAE(S) = 0.159s$, and $RMSE(S) = 0.126s$. And the MAE and RMSE of both P and S waves using DynaPicker are $MAE(P) = 0.127s$, $RMSE(P) = 0.128s$, and $MAE(S) = 0.198s$, and $RMSE(S) = 0.137s$. However, it is worth noting, that the original EQ-Transformer is trained on labeled arrival-time seismic data of the STEAD dataset, while DynaPicker is only trained on the short-window SCEDC dataset without phase arrival-time labeling.

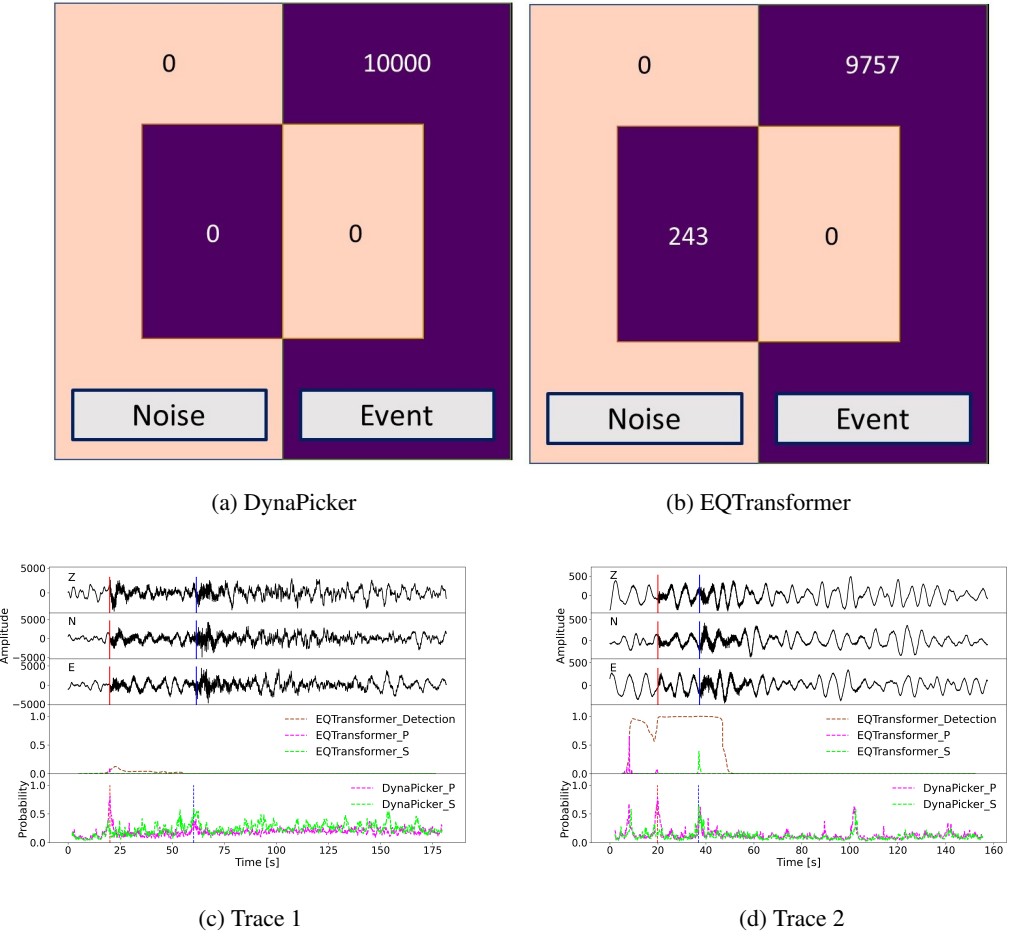

(a) DynaPicker

(b) EQTransformer

(c) Trace 1

(d) Trace 2

**Figure 5.** Visualizations of the testing result on the Iquique dataset. (a) and (b) are the confusion matrices from in-domain experiments for DynaPicker and EQTransformer, respectively. Here, the pre-trained model of EQTransformer is directly used without retraining and adopted from Seisbench (Woollam et al., 2022), where DynaPicker is able to detect all earthquake events compared with EQTransformer. (c) and (d) plots the EQTransformer and DynaPicker predictions on two waveform examples from the Iquique dataset. In (c) and (d), the upper three subplots are the three-component seismic waveforms where the vertical red and blue lines correspond to the ground truth arrival time of P- and S-phases from the dataset metadata, respectively, and the bottom subplots display the predicted probability for P- and S-phases by using EQTransformer and DynaPicker, respectively, where the dashed vertical lines in red and blue depict the locations of the maximal predicted probabilities of P- and S-phases, respectively. For EQTransformer, in (c) only the P-pick is detected at a low probability, whereas the S-pick is missing, and in (d) multiple picks are predicted, especially one incorrectly P-phase is detected at a high probability. For DynaPicker, both the true P- and S-phases are detected with a higher probability compared with EQTransformer.

## 5.5 Earthquake detection

In this subsection, we further test DynaPicker's performance in P-wave onset detection using the published CREIME model (Chakraborty et al., 2022b) for magnitude estimation. We selected varying time intervals of data recorded on February 6th, 2023 to test the detection of diverse aftershocks (See B1). The data corresponds to seismograms from stations that are part of the seismic network operated by the Kandilli Observatory and Earthquake Research Institute, known as KOERI (Kandilli Observatory And Earthquake Research Institute, Boğaziçi University, 1971). The information regarding arrival times, locations, and magnitude estimations was obtained from the catalog of the Bogazici University Kandilli Observatory and Earthquake Research Institute National Earthquake Monitoring Center.

We begin by feeding the aftershock waveform of the 2023 Turkey earthquake data into DynaPicker to obtain the P-phase probabilities for each sample. We then use both the waveform windows for which the P-phase probability exceeds 0.7 as input for the CREIME model to estimate the magnitude of the aftershocks. Finally, a seismological expert cross-validates the estimated magnitude with the Turkey earthquake catalog (see Table B1). The results of this analysis are presented in Figure 6. The waveform is first analyzed by Dynapicker. Subsequently, the windows for which Dynapicker P-pick probability is higher than 0.7 are fed to the CREIME Model for magnitude estimation. The magnitudes predicted by CREIME are shown by the red circles in Figure 6 and the catalog magnitudes are shown by the purple squares. A slight underestimation is observed which can be attributed to noise in the data and the use of different magnitude scales. This will be looked into in future works. A predicted magnitude less than -0.5 by CREIME represents noise. Figure 7 shows two earthquakes that had at least three stations within 1°. One is successfully detected at two stations while the other is only in one.

## 6 Discussion

### 6.1 Model retraining

We have also performed retraining on all the models, including DynaPicker, GPD, and CapsPhase, using the SCEDC dataset and applying the early stopping technique same as GPD (Ross et al., 2018) and CapsPhase (Saad and Chen, 2021). The SCEDC dataset was divided randomly into a training dataset (90%), a validation dataset (5%), and a testing dataset (5%). Besides, different from the original DynaPicker, here we employ a large kernel size and replace the first static convolution layer with the 1D-DCD block. The testing results for three models are as follows: DynaPicker (98.96%), GPD (98.85%), and CapsPhase (98.45%). The results indicate that incorporating a larger kernel size and replacing the initial static convolution layer with the 1D-DCD block in the original DynaPicker model indeed contribute to improving its performance, However, these modifications did not result in a significant enhancement for DynaPicker. Conversely, the accuracy of CapsPhase was observed to be lower compared to the reported findings in the original paper (Saad and Chen, 2021). Moreover, the retrained versions of DynaPicker and GPD exhibit a greater margin of error when it comes to seismic phase picking, in contrast to the original DynaPicker and the pre-trained GPD (Ross et al., 2018). Consequently, for the seismic phase picking task in this study, we will utilize the pre-trained models of GPD and CapsPhase.

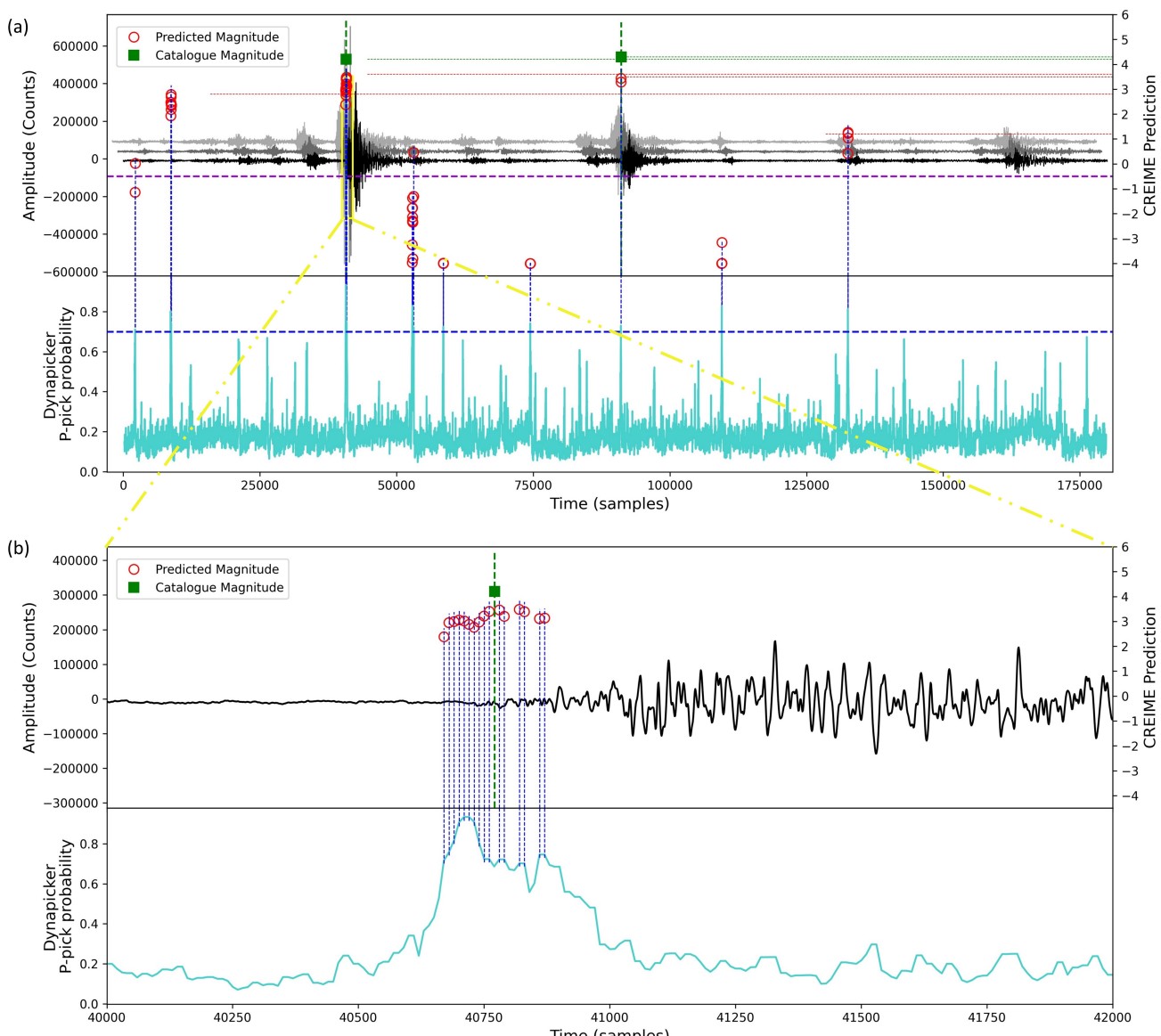

**Figure 6.** A visualization for combining Dynapicker and CREIME on a seismic recording. Dynapicker uses 3 component waveforms to output probabilities corresponding to P- and S-arrivals. The waveform windows with a P-pick probability higher than 0.7 are fed to the CREIME model for magnitude estimation. The red circles represent CREIME predictions while the green squares represent catalog magnitude. A CREIME prediction less than -0.5 (marked with purple dotted line in uppermost subfigure) represents noise. Please note that the time axis correlates with the Z-component of the seismogram, shown in black.

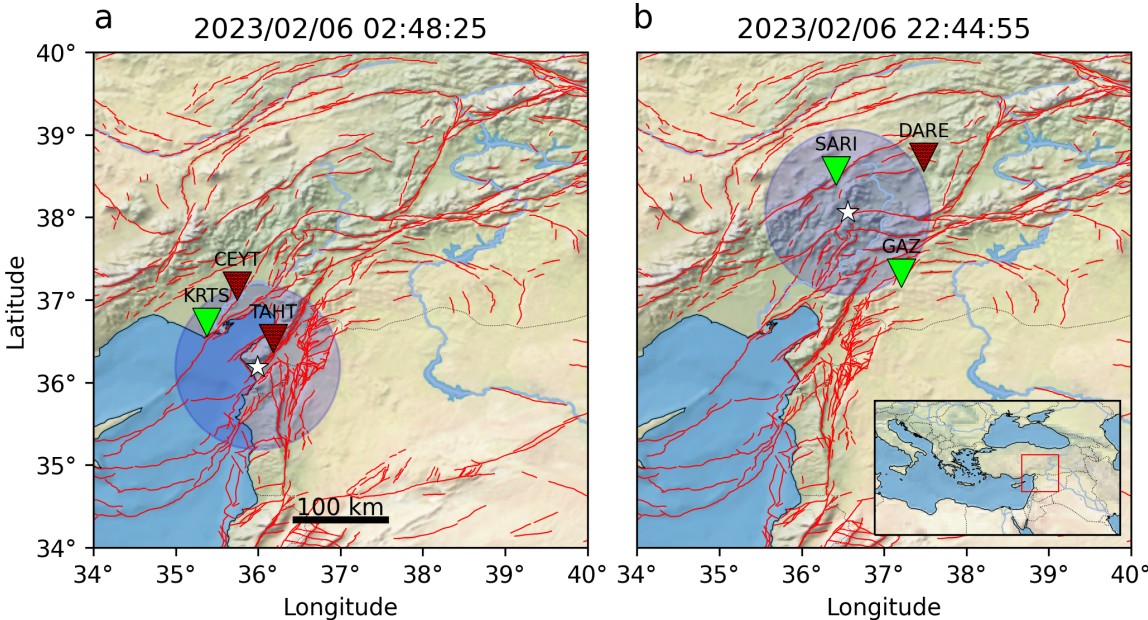

**Figure 7.** This figure shows two example earthquakes from the recent Turkey Earthquake series, and how they were detected using DynaPicker. Green stations correspond to a successful detection, and dotted red ones to an unsuccessful one. The earthquake epicenter is marked with a star, together with a $1°$ radius around it, the targeted range for DynaPicker. Additionally, we show the active faults in the region as taken from (Zelenin et al., 2022) with red lines.

### 6.2 Challenges

In Table B1, it has been noticed that in some cases, DynaPicker struggles to detect the phase pick for the live data of the Turkey earthquake. These challenges necessitate further investigation and improvement in future endeavors. Firstly, DynaPicker divides the continuous seismic record into 4-second overlapping windows, which means its detection performance depends on the arrival-time difference between P- and S- phases and the shifted numbers applied. Secondly, when evaluating the magnitude using CREIME, there seems to be an underestimation of the magnitude compared to the catalog values. This discrepancy

could be attributed to data noise or variations in magnitude scales utilized in the catalog. Last but not least, the fed data of DynaPicker is filtered by the bandpass filter, hence, the picking performance is contingent upon the quality of the seismic data. Our forthcoming work aims to address these challenges comprehensively.

### 7 Conclusions

This study first introduces a novel seismic phase classifier based on dynamic CNN, which is subsequently integrated into a

365 deep learning model for magnitude estimation. The classifier consists of a conventional convolution layer and multiple dynamic

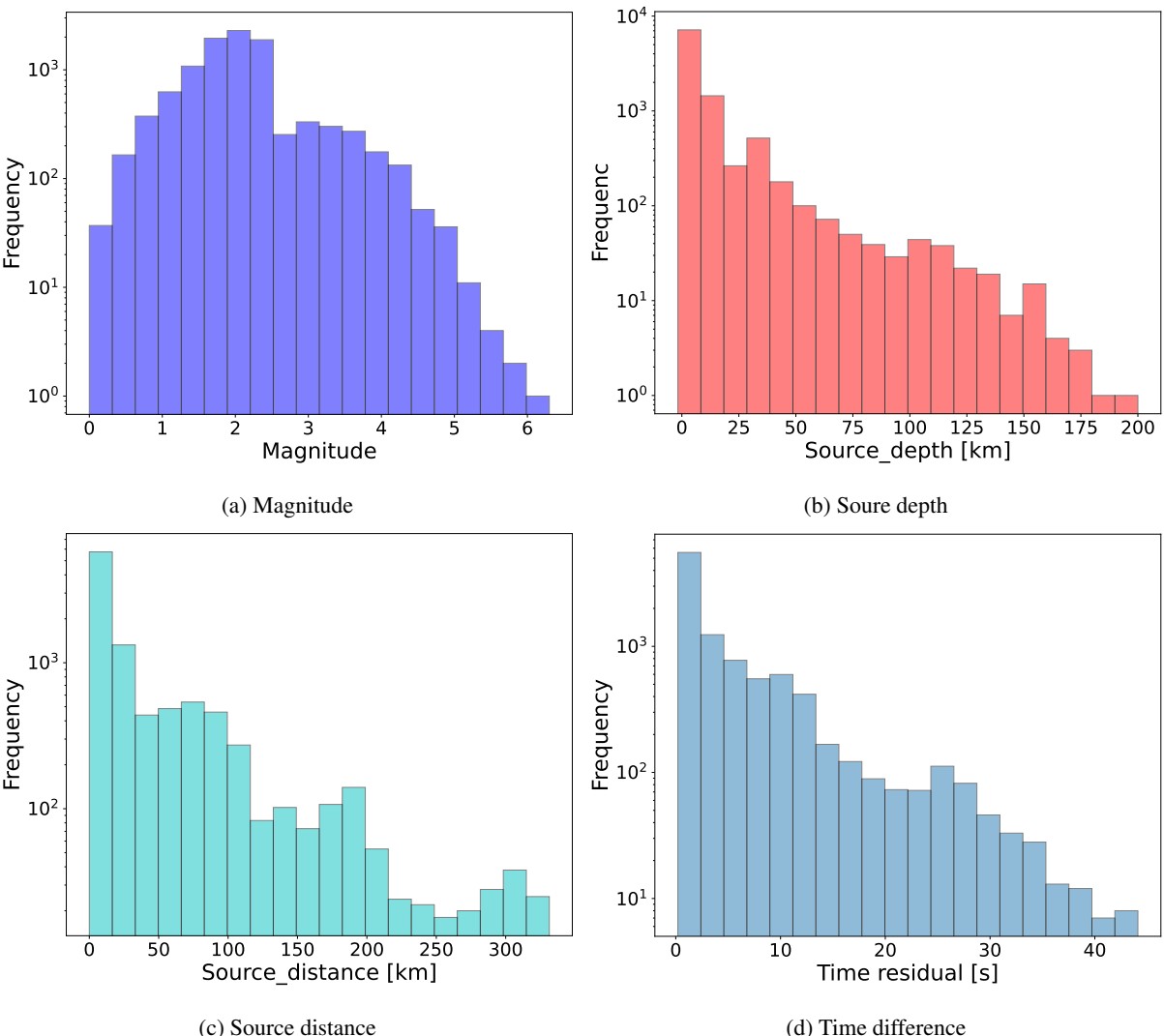

**Figure 8.** Distribution of (a) earthquake magnitudes, (b) earthquake source depths, (c) earthquake source distances, and (d) time difference between P-phase and S-phase arrival-time of the subset from the STEAD dataset (Mousavi et al., 2019a).

convolution decomposition layers. To train the proposed seismic phase classifier, we use seismic data collected by the Southern California Seismic Network. The classifier exhibits promising results during testing with earthquake waveforms recorded globally, demonstrating its good generalization ability. Extensive experiments demonstrate that this model yields superior performance over several baseline methods on phase identification and phase picking. The results from our work contribute 370 to the existing body of literature on supervised learning-based methods for seismic phase classification and demonstrate that with appropriate considerations regarding overfitting and generalization, such methods can improve seismological processing workflows, not just for large catalogs, but also for varying datasets.

**Table A1.** Body-wave arrival time evaluation using different temperatures on the STEAD dataset.

| $T$ | No. of undetected events | No. of abs(e) $\leq$ 0.5s for P-pick | $\mu_P$ | $\sigma_P$ | No. of abs(e) $\leq$ 0.5s for S-pick | $\mu_S$ | $\sigma_S$ |
|---|---|---|---|---|---|---|---|
| 1 | 0 | 8926 | 0.018 | 0.132 | 7168 | -0.003 | 0.199 |
| 4 | 0 | 9032 | 0.005 | 0.125 | 6984 | 0.002 | 0.196 |
| 10 | 0 | 9063 | 0.0008 | 0.123 | 6857 | 0.004 | 0.196 |
| 20 | 0 | 9084 | -0.001 | 0.122 | 6797 | 0.004 | 0.196 |

$\mu_P$ and $\sigma_P$ are the mean and standard deviation of errors (ground truth − prediction) in seconds respectively for P phase picking. $\mu_S$ and $\sigma_S$ are the mean and standard deviation of errors (ground truth − prediction) in seconds respectively for S phase picking.

*Data availability.* The seismic dataset of the Southern California Earthquake Data Center used in this study can be accessed at https://scedc. caltech.edu/data/deeplearning.html. The STEAD data can be downloaded from https://github.com/smousavi05/STEAD, and the INSTANCE dataset is freely available at http://www.pi.ingv.it/instance/. The details about how to download and use the Iquique dataset can be found in SeisBench (Woollam et al., 2022).

## Appendix A: Parameter investigation

In this part, the impact of different temperatures in the softmax function (see Eq. (5) for illustration.), and different shift numbers ($n_{shift}$) on the model performance of phase arrival-time picking for continuous seismic waves are investigated. Here, $1 \times 10^4$ samples are randomly selected from the STEAD dataset (Mousavi et al., 2019a). The distribution of earthquake magnitudes, earthquake source depth, earthquake source distance, and time difference between P-phase and S-phase arrival-time are displayed in Figure 8.

### A0.1 Impact of different temperatures.

In this part, the impact of different temperatures in the used softmax function on the model performance of phase arrival-time picking for the continuous seismic wave is investigated as summarised in Table A1, in which $n_{shift}$ is fixed as 10. From Table A1, in this work, the temperature $T$ is empirically set to 4.

### A0.2 Impact of shift numbers.

This part studies the effect of different shift numbers ($n_{shift}$) on seismic onset arrival time estimation, where the temperature $T$ is set to 4. From Table A2, we can conclude that the results of $n_{shift} = 5$ and $n_{shift} = 10$ are close, while according to Eq. (4), lower shift number increases the numbers of the sliding window, and more time is used to locate the arrival-time. Hence, in this study, the shift number $n_{shift}$ is set as 10.

**Table A2.** Body-wave arrival time evaluation using different shift numbers on the STEAD dataset.

| $n_{shift}$ | No. of undetected events | No. of abs(e) $\leq 0.5$s for P-pick | $\mu_P$ | $\sigma_P$ | No. of abs(e) $\leq 0.5$s for S-pick | $\mu_S$ | $\sigma_S$ |
|---|---|---|---|---|---|---|---|
| 5 | 0 | 9050 | -0.005 | 0.120 | 7032 | -0.004 | 0.195 |
| 10 | 0 | 9032 | 0.005 | 0.125 | 6984 | 0.002 | 0.196 |
| 20 | 0 | 8947 | 0.016 | 0.119 | 6975 | 0.003 | 0.201 |
| 100 | 0 | 8652 | 0.012 | 0.111 | 5803 | 0.024 | 0.248 |

$\mu_P$ and $\sigma_P$ are the mean and standard deviation of errors (ground truth − prediction) in seconds respectively for P phase picking. $\mu_S$ and $\sigma_S$ are the mean and standard deviation of errors (ground truth − prediction) in seconds respectively for S phase picking.

## Appendix B: Magnitude estimation by combining DynaPicker and CREIME on the aftershock sequence of Turkey earthquake.

*Author contributions.* Wei Li: Conceptualization, Methodology, Software, Writing - Original Draft, Writing - Review and Editing. Megha Chakraborty: Writing - Review and Editing, Visualization and Analysis. Jonas Köhler: Writing - Review and Editing, Visualization. Claudia Quinteros-Cartaya: Data analysis and Validation. Georg Rümpker: Writing - Review and Editing. Nishtha Srivastava: Conceptualization, Methodology, Writing - Review and Editing

*Competing interests.* The authors declare that they have no conflict of interest.

*Acknowledgements.* This work is supported by the "KI-Nachwuchswissenschaftlerinnen" - grant SAI 01IS20059 by the Bundesministerium für Bildung und Forschung - BMBF. Calculations were performed at the Frankfurt Institute for Advanced Studies' GPU cluster, funded by BMBF for the project Seismologie und Artifizielle Intelligenz (SAI). We thank the authors of Seisbench (Woollam et al., 2022) for their help to use the saved model of EQTransformer (Mousavi et al., 2020) in the Pytorch version, and also thank Dr. Omar M. Saad for his help in using the pre-trained model of CapsPhase (Saad and Chen, 2021). We also would like to thank Johannes Faber for his helpful discussion.

**Table B1.** Statistical results of magnitude estimation. Mag (MLv) and MagAv (MLv) are the individual magnitudes of each station and the magnitude average values from all stations, respectively, according to the KOERI-RETMC catalog (Boun Koeri Regional Earthquake-Tsunami Monitoring Center). $\mu_{mag}$ and $\sigma_{mag}$ are the mean and standard deviation of the magnitude calculated by CREIME for consecutive time windows for which the P-arrival probability calculated by Dynapicker exceeds the threshold of 0.7

| Station | Event_P_arrival_Time | Mag (MLv) | MagAv (MLv) | $\mu_{mag}$ | $\sigma_{mag}$ |
|---------|----------------------|-----------|-------------|-------------|----------------|
| SLFK    | 2023-02-06 02:35     | 4.05      | 4           | 0           |                |
| KRTS    | 2023-02-06 13:09     | 4.38      | 4.4         | 1.76        | 0.15           |
| KRTS    | 2023-02-06 13:36     | 3.39      | 4           | 2.68        | 0.14           |
| ARPRA   | 2023-02-06 03:46     | 5.3       | 4.9         | 3.08        | 0.12           |
| SARI    | 2023-02-06 22:45     | 3.73      | 3.9         | 3.21        | 0.13           |
| SARI    | 2023-02-06 23:21     | 4.07      | 3.6         | 3.47        | 0.3            |
| DARE    | 2023-02-06 22:45     | 3.63      | 3.9         | 0           |                |
| DARE    | 2023-02-06 22:55     | 3.66      | 4           | 0           |                |
| DARE    | 2023-02-06 23:02     | 3.32      | 3.4         | 3.34        | 0.1            |
| DARE    | 2023-02-06 23:21     | NO        | 3.6         | 0           |                |
| DARE    | 2023-02-06 23:56     | 3.8       | 3.7         | 3.42        | 0.24           |
| IKL     | 2023-02-06 05:44     | 2.77      | 3.4         | 3.19        | 0.21           |
| KRTS    | 2023-02-06 05:36     | 3.94      | 4.6         | 2.32        | 0.16           |
| TAHT    | 2023-02-06 02:23     | 5.6       | 5.3         | 0           |                |
| TAHT    | 2023-02-06 02:48     | 3.86      | 4.1         | 0           |                |
| URFA    | 2023-02-06 03:29     | 4.39      | 4.6         | 2.95        | 0.17           |
| KRTS    | 2023-02-06 02:23     | 4.96      | 5.3         | 2.85        | 0.11           |
| KRTS    | 2023-02-06 02:48     | 4.12      | 4.1         | 2.15        | 0.11           |
| GAZ     | 2023-02-06 13:18     | 5.29      | 5.2         | 4.44        | 0.31           |
| GAZ     | 2023-02-06 03:58     | 3.79      | 4.6         | 0           |                |
| CEYT    | 2023-02-06 05:36     | 4.24      | 4.6         | 3.43        | 0.27           |
| CEYT    | 2023-02-06 02:23     | 5.22      | 5.3         | 4.27        | 0.24           |
| CEYT    | 2023-02-06 02:48     | 3.9       | 4.1         | 0           |                |
| GAZ     | 2023-02-06 02:31     | 4.63      | 4.8         | 3.61        | 0.08           |
| GAZ     | 2023-02-06 22:42     | 3.57      | 3.9         | 3.27        | 0.18           |
| GAZ     | 2023-02-06 22:51     | 3.2       | 3.3         | 3.54        | 0.14           |
| GAZ     | 2023-02-06 22:55     | 4.3       | 4           | 3.34        | 0.31           |
| GAZ     | 2023-02-06 23:13     | 4.18      | 4.2         | 3.61        | 0.25           |
| GAZ     | 2023-02-06 23:27     | NO        | 3.6         | 3.17        | 0.62           |

$\mu_{mag}$ and $\sigma_{mag}$ are the mean and standard deviation of the estimated magnitude. Here, NO means there is magnitude data in the catalog.

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
