# Peer review of "Earthquake Monitoring using Deep Learning with a Case Study on the Kahramanmaras Turkey Earthquake Aftershock Sequence"

_EGUsphere, 2023_

## Author Comment (AC1)

**Response to Reviewers**

**Dear Reviewer,**

We appreciate the time and effort that you have dedicated and are grateful for your insightful comments which have improved the manuscript. We have incorporated the constructive suggestions, and have highlighted the changes within the manuscript and marked them in blue color. Here is a point-by-point response to your comments and concerns.

**Reviewer 2**

The manuscript proposes a new deep-learning picker that leverages dynamic convolutional neural networks for detecting and picking seismic phases from windowed or continuous waveform data. The authors then combined the previously published CREIME model for magnitude estimations of waveform windows that have high P-wave probabilities. The authors have evaluated the performance of their picker and their combined workflow on open-source seismic datasets and aftershocks following the Turkey earthquake. The technical part of the manuscript is overall solid. However, I have vital concerns about the 'real-time' claim. It seems to me that the authors have confused the concept of processing continuous data with the concept of real-time earthquake monitoring. I suggest the authors modify their claim from 'Real-time' to 'efficient' and emphasize more on the performance of the proposed deep-learning picker. Aside from the 'real-time' claim, the study seems good overall. Below are my detailed comments:

**Response:** We appreciate the reviewer's suggestion to modify our claim from 'real-time' to 'efficient'. We understand that 'real-time' can have different interpretations, and we agree that emphasizing the efficiency of our proposed deep-learning picker is essential. We made this adjustment in the revised manuscript avoiding the strict interpretation of 'real-time'.

1.How is the term 'Real-time' defined? What is the time cost between the time of data recorded at the seismometer and the time of output produced? Please note that there are several important steps for real-time earthquake monitoring besides the time cost of the phase-picking model. For example, how is the time cost of the data transmitted from the seismometer to the data center? Is the data processed at the seismometer end with edge-computing (which would be important in areas with poor internet access), or is the data transmitted to the data center first and processed later there? The data packages in the real-time seismic data flow can contain errors due to transmission issues. How is that addressed?

**Response:** We acknowledge that the term 'real-time' to describe our model's performance was a misnomer. Our model does not operate in real-time, and therefore, we did not perform any analysis on the time cost of real-time data flow. We apologize

for any confusion caused by the incorrect terminology and thank the reviewer for pointing this out. Instead, our proposed model provides timely results from continuous waveform recordings. We revised the manuscript to accurately reflect this and ensure that our terminology aligns with the actual capabilities of our model.

2. What is the inference time cost of the model? What is the key advantage of the proposed method over conventional and lightweight convolutional deep-learning pickers in terms of real-time monitoring? The authors claim, "However, most of the prevalent CNN-based models perform inference using static convolution kernels, which may limit their representation power, efficiency, and ability for interpretation." However, to my acknowledgment, the current CNN-based models, especially lightweight ones, are sufficient for millisecond-level inference. One key claim of the manuscript is that the proposed method is much faster and, therefore, more suitable for real-time earthquake monitoring. However, I didn't find any quantitative comparisons on the inference speed in this paper.

**Response:** In the introduction section of the manuscript, we claim that "However, most of the prevalent CNN-based models perform inference using static convolution kernels, which may limit their representation power, efficiency, and ability for interpretation." To clarify, our primary focus in this work is the utilization of the dynamic networks to enhance the performance of the seismic phase classification performance and, consequently reduce the errors in the phase arrival-time estimation. As a result, we have not quantified the time comparison of the inference speed in this paper. However, following the reviewer's suggestion, we will explore such a relative comparison in our follow-up work.

3. The event's location is one key information in earthquake monitoring and yet not resolved by the current workflow. The lack of event location information would decrease the significance of the proposed monitoring method.

**Response:** Event location is indeed a crucial component of seismic analysis, but its inclusion in our current model was beyond the scope of this work. We understand the importance of this aspect and recognize it as an essential feature for a comprehensive seismic monitoring system. However, getting an accurate phase arrival is crucial for a correct event location determination. We plan to address event location information and upgrade the current model in future research.

4. Why is being adaptive to different input lengths important? Is that because in the real-time earthquake monitoring scenario that the authors are dealing with, the input lengths of data chunks can be significantly different? And what are the advantages of the proposed method over the RNN-based pickers, which can also adapt to different input lengths?

**Response:**

- The proposed method is adaptive to different input lengths, as it can accommodate the continuous data of different durations. We have edited the text in line 40 to highlight this.

- We are currently working on a RNN-based model for event detection which will be a follow-up of this work.

5. Section 5.5 'Real-time earthquake detection', how is the 'real-time' here different from 'continuous data'? Section 6.2 'the live data of the Turkey earthquake', what does the 'live data' mean, do authors have access to the real-time data packages from the Earthquake Data Center System of Turkey, or do they use the downloaded continuous waveform data?

**Response:** We incorrectly used the term 'real-time' when referring to our data source. we apologize again for the inconvenience. To accurately describe our data source, we now use the term 'continuous seismic recordings' throughout the revised manuscript. This term better reflects the downloaded continuous waveform data that we are utilizing.

---

## Author Comment (AC2)

**Response to Reviewers**

**Dear Reviewer,**

We appreciate the time and effort that you have dedicated and are grateful for your insightful comments which have improved the manuscript. We have incorporated the constructive suggestions, and have highlighted the changes within the manuscript and marked them in blue color. Here is a point-by-point response to your comments and concerns.

**Reviewer 1**

In this work, the authors apply the dynamic convolutional neural network to two tasks: seismic phase classification and arrival-time picking. They compared the new model, DynaPicker, to a few other deep learning models and demonstrated that DynaPicker could achieve better performance for input data of different lengths. The main concern I have for this work is that the model comparison may not be very accurate. The reported improvements in precision/recall/F1 scores are not significant, so the performance of DynaPicker may become even worse if choosing a slightly different threshold. Based on the selected examples shown in the paper, the false positive rate of the new model could be very high. I would request the authors to plot precision-recall curves to compare the performance of different models to avoid bias in selecting a specific threshold for comparison. One good example is Münchmeyer et al.'s work of "Which Picker Fits My Data?"

**Response:** We thank the reviewer for the constructive suggestion.

- Following Münchmeyer et al.'s work "Which Picker Fits My Data?", we have plotted the receiver operating characteristic (ROC) curves for the DynaPicker and GPD models as follows. Based on Figure 1, it's evident that DynaPicker exhibits a similar low false positive rate to the GPD model. Furthermore, the picking error distribution summarized in Tables 4 and 5 that DynaPicker performs better in phase arrival time picking than GPD.

- Unfortunately, even with our multiple attempts, the CapsPhase model retrieved from the git repository cannot be utilized at this time. When loading the CapsPhase model in the created virtual environment, we received a segmentation fault error, even after increasing the memory size. We will keep on trying to perform the comparison with the CapsPhase model and will address this in our follow-up work.

1. Table 4: I have three comments of the reported results: (1) Based on the standard deviations of the time residuals, we can see clearly DynaPicker has a very large time error for both P and S phases. I am wondering if DynaPicker is really an improved

(a) DynaPicker

(b) GPD

**Figure 1.** ROC curves

alternative to current models. (2) In the table, only the number of picks ($< 0.5$s) is reported. But how many picks ($> 0.5$s) does DynaPicker detected? It is important to report the false positives. (3) The absolute number of undetected events is not helpful. What activation threshold do you use? How many false positive events do you detected in order to detect all true events?

**Response:**

– Here we followed the CapsPhase work definition i.e., 'if the error between the model's predicted picks and the ground truth picks have an absolute error below 0.5s, then it is true positive'. As indicated in Table 4, the time residuals of DynaPicker exhibit standard deviations that are either similar to or smaller than those of other models. In certain instances, DynaPicker even surpasses the other models by having lower standard deviations.

– In Table 4, a total number of 10,000 events for each scenario are randomly selected from the STEAD dataset. All of these earthquake events are correctly detected using DynaPicker. In case 1 (used model: DynaPicker), there are 945 events with P-phase picking errors exceeding 0.5s and 2304 events with S-phase picking errors exceeding 0.5s, respectively. In case 2 (used model: GPD), there are 2595 events with P-phase picking error and 2403 events with S-phase picking error greater than 0.5s, respectively. In both Tables 4 and 5, we have introduced two additional columns to denote the number of picks exceeding 0.5s for both P- and S-phases as shown in the following tables.

– In the task of the seismic phase classification with the SCEDC dataset, we did not use any activation threshold for event detection. However, in the context of analyzing the aftershock sequence of the Turkey earthquake, we empirically established an activation threshold of 0.7 for detecting events. We would further like to point out that the threshold should indeed be chosen carefully by the user based on the station and the data and what we show here is an example. The threshold of 0.7 was chosen experimentally to get the most optimum balance between false positives and false negatives.

**Changes in manuscript:** We have updated Tables 4 and 5 in the manuscript. Line 78-80 are added

2. Fig. 2: I am confused by this plot. If the predicted scores are also pretty high for waveforms that are not P or S phases, there could be many false positives. Based on the examples shown in Fig. 5, we can see DynaPicker can also easily pick up false positives.

**Response:** Figure 2 provides a schematic representation of the arrival time picking process for continuous seismic data, employing various window sizes while processing the same continuous waveform. Even though the probability might be relatively high ($> 0.7$) in few other windows, we opt for the window with the highest P/S probability, which usually is in the order of 0.99, to estimate the phase arrival time. As illustrated in the initial ROC curves, by following this approach, DynaPicker exhibits a minimal number of false positives.

3. Eq. 5: Did you compare the results using $T = 1$ and $T = 4$ for the phase picking problem? Because the temperature softmax function is not used by previous works of phase picking, it is necessary to demonstrate that it can help the phase picking task.

**Response:** We concur with this observation of the reviewer. We have summarized the outcomes of utilizing different temperatures for phase picking in the Appendix. The relevant table is presented below. You can find this table on page 22 of the

**Table 4.** Body-wave arrival time evaluation using different methods on STEAD dataset including (a) $S_{arrival} - P_{arrival} > 4s$ and (b) $S_{arrival} - P_{arrival} < 4s$. In each case, $1 \times 10^4$ samples are used. Same as the CapsPhase paper, the event whose pick predicted by a model has an absolute error larger than 0.5 s, is recognized as false positive.

(a) $S_{arrival} - P_{arrival} > 4s$

| Method | No. of events detected | No. of abs(e) ≤ 0.5s for P-pick | $\mu_P$ | $\sigma_P$ | No. of abs(e) > 0.5s for P-pick | No. of abs(e) ≤ 0.5s for S-pick | $\mu_S$ | $\sigma_S$ | No. of abs(e) > 0.5s for S-pick |
|---|---|---|---|---|---|---|---|---|---|
| DynaPicker | 10000 | **9055** | **0.0002** | 0.151 | 945 | **7696** | **0.011** | 0.203 | 2304 |
| GPD | 9826 | 8975 | -0.0036 | 0.149 | 851 | 2623 | -0.043 | 0.193 | 7203 |
| CapsPhase | 9885 | 8766 | -0.018 | 0.149 | 1119 | 5545 | -0.112 | 0.184 | 4340 |
| AR picker | 10000 | 7963 | 0.079 | 0.133 | 2037 | 4011 | 0.205 | 0.176 | 5989 |

(b) $S_{arrival} - P_{arrival} < 4s$

| Method | No. of events detected | No. of abs(e) ≤ 0.5s for P-pick | $\mu_P$ | $\sigma_P$ | No. of abs(e) > 0.5s for P-pick | No. of abs(e) ≤ 0.5s for S-pick | $\mu_S$ | $\sigma_S$ | No. of abs(e) > 0.5s for S-pick |
|---|---|---|---|---|---|---|---|---|---|
| DynaPicker | 10000 | **9405** | **0.0048** | 0.091 | 595 | **7597** | **0.0075** | 0.179 | 2403 |
| GPD | 9662 | 8890 | 0.0059 | 0.092 | 772 | 4393 | -0.012 | 0.164 | 5269 |
| CapsPhase | 9861 | 8767 | -0.020 | 0.084 | 1094 | 5545 | -0.061 | 0.164 | 4316 |
| AR picker | 10000 | 7755 | 0.015 | 0.075 | 2245 | 7369 | 0.126 | 0.161 | 2361 |

$\mu_P$ and $\sigma_P$ are the mean and standard deviation of errors (ground truth − prediction) in seconds respectively for P phase picking. $\mu_S$ and $\sigma_S$ are the mean and standard deviation of errors (ground truth − prediction) in seconds respectively for S phase picking.

manuscript.

4. L230: "we can observe that EPick achieves the best performance in phase picking over DynaPicker by using different window sizes." Does this mean the claimed advantage of DynaPicker for different input length is not true? Although you explain that the reason is that EPick is pre-trained using the STEAD dataset, you can also train DynaPicker using the STEAD dataset to make the comparison more accurate.

**Response:** In response to the reviewer's recommendation, we proceeded to retrain the EPick model using the same dataset sourced from the STEAD data, which serves as the training data for DynaPicker. Subsequently, we employed the retrained EPick model to estimate phase arrival times for continuous data extracted from the STEAD dataset. It is important to highlight that there is no overlap between the datasets used for EPick training and those utilized for phase arrival time detection. We observed that, while the performance in detecting the P phase was similar, the accuracy of S-Phase picking decreased from a mean value of -0.002s to -0.050s, and the standard deviation increased from 0.122s to 0.147s. Additionally, The EPick model is developed for the task of estimating seismic phase arrival times with fixed input length. In contrast, the DynaPicker model

**Table 5.** Body-wave arrival time evaluation using different methods on INSTANCE dataset including (a) $S_{arrival} - P_{arrival} > 4s$ and (b) $S_{arrival} - P_{arrival} < 4s$. In each case, $1 \times 10^4$ samples are used. Same as the CapsPhase paper, the event whose pick predicted by a model has an absolute error larger than 0.5 s, is recognized as false positive.

(a) $S_{arrival} - P_{arrival} > 4s$

| Method | No. of events detected | No. of abs(e) ≤ 0.5s for P-pick | $\mu_P$ | $\sigma_P$ | No. of abs(e) > 0.5s for P-pick | No. of abs(e) ≤ 0.5s for S-pick | $\mu_S$ | $\sigma_S$ | No. of abs(e) > 0.5s for S-pick |
|---|---|---|---|---|---|---|---|---|---|
| DynaPicker | 10000 | **8707** | 0.030 | 0.130 | 1293 | **7530** | **0.019** | 0.199 | 2470 |
| GPD | 9623 | 8231 | 0.028 | 0.123 | 1392 | 4726 | -0.032 | 0.179 | 4897 |
| CapsPhase | 9598 | 7948 | 0.014 | 0.140 | 1650 | 5837 | -0.103 | 0.186 | 3761 |
| AR picker | 9999 | 7545 | 0.052 | 0.118 | 2454 | 3274 | 0.218 | 0.168 | 6725 |

(b) $S_{arrival} - P_{arrival} < 4s$

| Method | No. of events detected | No. of abs(e) ≤ 0.5s for P-pick | $\mu_P$ | $\sigma_P$ | No. of abs(e) > 0.5s for P-pick | No. of abs(e) ≤ 0.5s for S-pick | $\mu_S$ | $\sigma_S$ | No. of abs(e) > 0.5s for S-pick |
|---|---|---|---|---|---|---|---|---|---|
| DynaPicker | 10000 | **8690** | **0.012** | 0.079 | 1310 | **7815** | **0.0085** | 0.160 | 2185 |
| GPD | 9833 | 8109 | 0.022 | 0.075 | 1724 | 6647 | -0.019 | 0.134 | 3186 |
| CapsPhase | 9872 | 7984 | 0.019 | 0.091 | 1888 | 5447 | -0.072 | 0.143 | 4325 |
| AR picker | 10000 | 8296 | 0.016 | 0.077 | 1704 | 5778 | 0.149 | 0.168 | 4222 |

$\mu_P$ and $\sigma_P$ are the mean and standard deviation of errors (ground truth − prediction) in seconds respectively for P phase picking. $\mu_S$ and $\sigma_S$ are the mean and standard deviation of errors (ground truth − prediction) in seconds respectively for S phase picking.

**Table A1.** Body-wave arrival time evaluation using different temperatures on the STEAD dataset.

| $T$ | No. of undetected events | No. of abs(e) ≤ 0.5s for P-pick | $\mu_P$ | $\sigma_P$ | No. of abs(e) ≤ 0.5s for S-pick | $\mu_S$ | $\sigma_S$ |
|---|---|---|---|---|---|---|---|
| 1 | 0 | 8926 | 0.018 | 0.132 | 7168 | -0.003 | 0.199 |
| 4 | 0 | 9032 | 0.005 | 0.125 | 6984 | 0.002 | 0.196 |
| 10 | 0 | 9063 | 0.0008 | 0.123 | 6857 | 0.004 | 0.196 |
| 20 | 0 | 9084 | -0.001 | 0.122 | 6797 | 0.004 | 0.196 |

$\mu_P$ and $\sigma_P$ are the mean and standard deviation of errors (ground truth − prediction) in seconds respectively for P phase picking. $\mu_S$ and $\sigma_S$ are the mean and standard deviation of errors (ground truth − prediction) in seconds respectively for S phase picking.

was primarily designed for phase classification and its adaptation for phase arrival time detection with different input lengths

is a notable application. Furthermore, as the size of the training data increased, DynaPicker exhibited improved performance and demonstrated greater robustness when compared to EPick.

**Changes in manuscript:** Texts updated to help readers understand the process on pages 11 and 12.

5. L240: "The testing accuracy of DynaPicker is 98.82%, which is slightly greater than CapsPhase [30] (98.66%) and 1D-ResNet [9] (98.66%)." Because the differences are very small and do not tell readers much information, could you compare the waveforms of false predictions of these models to help understand where DynaPicker can be better?

**Response:** As per reviewer's suggestion, here, we plot several waveforms of false predictions, while they are correctly identified by DynaPicker.

[Figure]

**Figure 2.** Visualization of trace examples.

From these figures, we can observe that compared with other models, DynaPicker shows its advantage in phase classification. in scenarios where the ground truth label is noise and the seismic waveform exhibits increased noise levels, DynaPicker accurately identifies it as noise, whereas the GPD and ResNet models tend to misclassify it. Unfortunately, as mentioned before, the CapsPhase model retrieved from the git repository cannot be utilized at this time.

---

## Author Response (AR2)

**Response to Reviewers**

**Dear Reviewers,**

We appreciate the time and effort that you have dedicated and are grateful for your insightful comments which have improved the manuscript. We have incorporated the constructive suggestions, and have highlighted the changes within the manuscript and marked them in blue color. Here is a point-by-point response to your comments and concerns.

**Reviewer 1**

The authors have addressed most of my comments in the revision. However, I am still concerned about the model's performance.

I am unclear why the authors do not follow Figure 1 from "Which Picker Fits My Data?" to compare the model performance. The plot of the ROC curve is a very clear way to compare different models. My concern is that the model performance is actually worse.

The figure provided in the response file is also not very clear. Which dataset do the authors use to produce the plot? Please also plot the curves of models like GPD, PhaseNet, EqT, etc. in the same plot. If all models have a high true positive rate on your dataset, please zoom in on the range between like 0.9-1 for a clear comparison.

**Response:** We thank the reviewer for the constructive suggestion. In accordance with the Figure 1 presented in Münchmeyer et al.'s study titled "Which Picker Fits My Data?", we have generated receiver operating characteristic (ROC) curves for the P/S phases of the DynaPicker and GPD models using the SCEDC dataset, as depicted in Figure 1. This figure includes both a zoomed-in view within the range of 0.9-1 and the complete ROC curves. From Figure 1, we can observe that in comparison to GPD, the DynaPicker exhibits (a) a high true positive rate and (b) a slightly superior F1-score for P/S phase identification. Additionally, it is noteworthy that both the PhaseNET and EqT models are trained on the data with known ground truth arrival time for the P/S phases. However, since the SCEDC dataset lacks ground truth phase arrival times, we refrain from presenting ROC curves for these models in this context.

**Reviewer 2**

The manuscript proposes a new deep-learning picker that leverages dynamic convolutional neural networks for detecting and picking seismic phases from windowed or continuous waveform data. The authors then combined the previously published

(a) ROC curve of P-phase.

(b) ROC curve of S-phase.

**Figure 1.** Receiver operating characteristics for detection results from in-domain experiments for P-phase. The right subfigure allows us to assess the full curves of the performance evaluated on the SCEDC dataset. The left subfig shows zoomed-in parts of the upper left corner, with the zoom level to allow distinguishing the different models. Models were selected to maximize the Area Under Curve (AUC) score. Numbers in the corners indicate the test AUC scores. Markers indicate the point with the configuration associated with the highest F1 score.

CREIME model for magnitude estimations of waveform windows that have high P-wave probabilities. The authors have evaluated the performance of their picker and their combined workflow on open-source seismic datasets and aftershocks following the Turkey earthquake. The technical part of the manuscript is overall solid. The authors also corrected some previously raised issues. I think this manuscript is worthy of documentation at SE. I have only minor concerns regarding the motivation behind the proposed method and its application to the Kahramanmaras aftershock sequence. Below are my detailed comments:

The authors stated that 'most of the prevalent CNN-based models perform inference using static convolution kernels, which may limit their representation power, efficiency, and ability for interpretation.' However, the reported improvements are not that significant (from 98% to 99%). Can the authors elaborate on how these limitations and differences would affect the application of these models?

**Response:** In the context of phase classification, the DynaPicker model did not show a superior performance improvement. However, its application to the task of phase arrival time picking yielded noteworthy results. Through extensive testing on the continuous data sourced from multiple datasets, DynaPicker demonstrated an enhanced ability to detect earthquake events, showcasing robust and reliable performance in this specific application. Additionally, the DynaPicker performs better in phase classification when confronted with noisy data (as shown in the plots including the previous response).

Hence, this capability is indicative of the DynaPicker's resilience in capturing intricate features and patterns. In contrast, models relying on static convolution kernels exhibit limitations and differences that hinder their ability to capture the nuanced delineation of features and patterns.

2. The manuscript covers the Kahramanmaras aftershock too briefly. The entire article only has Section 5.5 on this topic. The authors may consider weakening the emphasis on the application of the Kahramanmaras aftershock in the title, as this part only occupies a small portion of the paper. Moreover, it lacks the phase association and event location using P and S phases from multiple seismic stations. Since the focus is on earthquake monitoring, earthquake location information is crucial. Besides, Section 6.2, it's not 'the live data of the Turkey earthquake'. You are using downloaded continuous waveforms, which is different from the live data stream.

**Response:** We express our gratitude for the reviewer's valuable suggestions.

– The application of the Kahramanmaras aftershock constitutes a minor portion of the paper, we have consequently revised the manuscript title.

– The significance of phase association and event location utilizing P and S phases from multiple seismic stations play a key role in the practical application of earthquake monitoring. We intend to dip into these aspects in our upcoming work.

– Within the manuscript, we have substituted the term "the live data of the Turkey earthquake" with "the continuous waveforms of the Turkey earthquake" in Section 6.2.